# EMMOE: A Comprehensive Benchmark for Embodied Mobile Manipulation in Open Environments

## Abstract

Developing autonomous household robots controlled by natural language has long been a pursuit of humanity. While advancements in large language models (LLMs) and embodied intelligence make this goal closer, several challenges persist: the lack of a robotic task and benchmark that are well-aligned with realistic household tasks, limited evaluation methods and metrics, data incompatibility between LLMs and mobile manipulation trajectories. To address these issues, we propose **E**mbodied **M**obile **M**anipulation in **O**pen **E**nvironments (EMMOE), a benchmark that requires agents to interpret user instructions and execute long-horizon everyday tasks in continuous space. EMMOE seamlessly integrates high-level and low-level embodied tasks into a unified framework, along with three new metrics for more diverse assessment. Additionally, we collect EMMOE-100, which features in various task attributes, detailed process annotations, re-plans after failures, and two sub-datasets for LLM training. Furthermore, we design HOMIEBOT, a sophisticated agent system consists of LLM with Direct Preference Optimization (DPO), light weighted navigation and manipulation models, and multiple error detection mechanisms. Finally, we demonstrate HOMIEBOT's performance and evaluations of different models and policies.

## 1 Introduction

Developing autonomous household robots capable of performing various daily tasks through a single instruction has been a long-standing goal. To achieve this goal, robots need to understand natural language instructions, make feasible plans, perceive and interact with dynamic environments, and equip with powerful navigation and manipulation skills. Typical methods like imitation learning (IL) (Ho & Ermon, 2016) and reinforcement learning (RL) (Sutton, 2018) primarily focus on task-specific policies, but are always limited to short-horizon tasks and struggle to generalize to new tasks. Task and Motion Planning (TAMP) treats long-horizon mobile manipulation tasks as hybrid discrete-continuous search problems (Garrett et al., 2021) and addresses with a hierarchical architecture (Kaelbling & Lozano-Pérez, 2011): High-level task planning in discrete task space, low-level motion planning in continuous action space, and interleaved execution between two layers. However, the scope of TAMP remains limited. Despite various extensions (Garrett et al., 2020; Chen et al., 2024), it still requires specific goal states and detailed scene configurations. The complexity and dynamism of real-world environments, and vague user descriptions make it highly challenging to meet these requirements.

In recent years, the rapid development of LLM (Achiam et al., 2023; DeepSeek-AI et al., 2025) and embodied intelligence (Brohan et al., 2023b; Driess et al., 2023) has made this pursuit possible. The scope of each layer in TAMP has been largely broadened and spawns various embodied tasks driven by language and vision. In high-level embodied tasks (Wu et al., 2023; Li et al., 2024a), LLMs have shown exceptional performance and powerful generalization capabilities. Advanced prompting techniques like Chain-of-Thought (COT) (Wei et al., 2022) have further enhanced the logical reasoning abilities of LLMs. Visual Language Models (VLMs) (Radford et al., 2021) enable agents to process visual inputs and understand current environments. Large Multi-modal Models (LMMs) (Liu et al., 2024) extend the application of embodied agents to real-world scenarios. The most recent world models (Matsuo et al., 2022) and spatial models (Huang et al., 2024b) allow agents to more accurately perceive scene information and spatial relationships. In low-level embodied tasks, the emphasis of models has gradually shifted from single skill with specific objects (Shafiullah et al., 2023) to single skill with

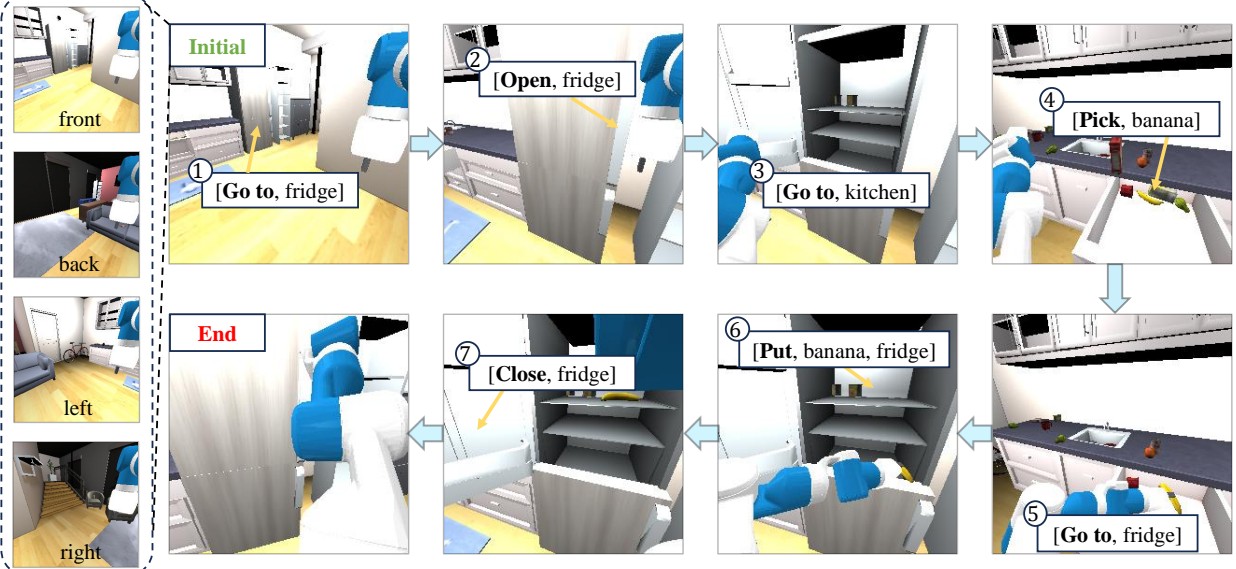

Figure 1: **Data example in EMMOE-100**. A key feature of EMMOE-100 is the emphasis on the reasoning process and interleaved execution. In the shown task, the agent must check the fridge first. Otherwise, even if the agent finally gets a banana in the kitchen, it will not be considered as a success.

open-vocabulary objects (Fang et al., 2023b), then further to general models (Black et al., 2024), such as Visual Language Navigation (VLN) (Zhang et al., 2024e) and Visual Language Action (VLA) (Brohan et al., 2023a) models.

However, several problems remain unresolved: 1) **Lack of a comprehensive task and benchmark aligns well with household scenarios.** Although significant progress has been made in various embodied tasks, there is still a gap between current tasks and envisioned language-driven household robots. Meanwhile, existing embodied tasks always operate in isolation, neglecting the mutual influence caused by interleaved task execution. By integrating different high-level and low-level embodied tasks, robots can achieve more advanced capabilities while enabling a unified evaluation of various embodied tasks. Each layer will constrain and influence the others, working collaboratively to accomplish the final task. 2) **Inadequate evaluation methods and metrics.** Embodied task planning involves causal dependencies between each step. However, solutions are not absolute, thus making evaluations based solely on individual steps or the final state insufficient. Additionally, current evaluation methods rely heavily on simulators or PDDL files, which also limits the real-world deployment and application. Furthermore, how to make more fine-grained evaluations of the entire agent system remains a challenge. 3) **LLM grounding problems.** Although LLMs excel in commonsense reasoning, they need to be grounded in current environments to produce realistic and practical outputs. Furthermore, due to the uncertainties and dynamic changes in the real world, LLMs must be able to make timely adjustments based on real-time feedback. However, the incompatibility between the conversation data required for LLMs and the trajectory data required for robotics increases the difficulty of grounding.

To advance the development of autonomous household robots, we propose EMMOE as an open challenge, which requires agents to interpret user instructions and execute long-horizon everyday tasks in continuous space. Besides, we manually collect EMMOE-100, the first daily task dataset featuring various task attributes, detailed process annotations, analyses of each output, re-plans after failures. We also build Supervised Fine-Tuning (SFT) and Direct Preference Optimization (DPO) (Rafailov et al., 2024) sub-datasets to facilitate the

alignment of LMM capabilities with specific embodied tasks. Finally, we introduce HOMIEBOT, a sophisticated agent system that integrates both high-level and low-level models, as well as multiple error detection and adaptation mechanisms. An example of EMMOE challenge and EMMOE-100 tasks is shown in Fig.1.

In particular, our paper makes the following contributions:

- We propose EMMOE, the first unified benchmark for both high-level and low-level embodied tasks with three novel metrics for more advanced evaluation.
- We collect EMMOE-100, the first everyday task dataset featuring COT outputs, diverse task designs, re-plan processes, with SFT and DPO sub-datasets.
- We design HOMIEBOT, a sophisticated agent system which integrates models at different levels, multiple error detection and adaptation mechanisms.

## 2 Related Work

### 2.1 Embodied Tasks and Benchmarks

With the rapid advancement of embodied agents and LLMs, diverse embodied tasks and benchmarks have emerged. Embodied Question Answering (EQA) (Das et al., 2018) requires agents to provide reasonable answers based on the current context, and evaluates the logical reasoning and visual perception abilities of LLMs (Majumdar et al., 2024; Dorbala et al., 2024). VLN needs agents to determine positions or directions based on visual observations and task instructions. It assesses the spatial understanding ability across diverse settings, including households (Shridhar et al., 2020), unseen environments (Yenamandra et al., 2023), and complex 3D spaces (Zhang et al., 2024b). Furthermore, manipulation datasets and benchmarks encompass table-top (Gu et al., 2023; Walke et al., 2023) and mobile manipulation (Fang et al., 2023a; O'Neill et al., 2024), and the pipeline ranges from traditional IL and RL (Fu et al., 2024b) to VLA models (contributors, 2024). Despite numerous benchmarks, a unified benchmark and task is still missing. Traditional mobile manipulation uses IL to learn entire trajectories, complicating the evaluation of intermediate processes. We propose EMMOE, a holistic benchmark designed to assess both final results and the execution process.

### 2.2 LLMs for Task Planning

Typical usages of LLM for task planning include treating LLM as a translator or a planner. Some studies integrate LLMs with traditional PDDL solvers (Guan et al., 2023; Zhou et al., 2024), treating the LLM as a translator between the real world and a specific domain solver. However, this paradigm is bottlenecked by the performance of the solver and fails to fully leverage the LLM's robust commonsense reasoning capabilities. When the LLM functions directly as a planner (Song et al., 2023), discrepancies between the model's outputs and physical reality frequently lead to execution failures. Although self-improvement techniques (Madaan et al., 2024; Shinn et al., 2024) can mitigate this problem, they rely heavily on prompt engineering and the model's intrinsic abilities. Besides, LLMs often struggle to self-correct when facing errors unrelated to planning. Furthermore, LLMs tend to repeat mistakes in similar scenarios as model weights remain unchanged. While RL-based adaptation (Ahn et al., 2022; Yao et al., 2023) mechanisms can adjust actions before execution, designing and training effective value functions or reward models is highly challenging. The DPO (Rafailov et al., 2024) algorithm greatly simplifies this process by requiring only a preference dataset. In our HOMIEBOT, we use DPO for model alignment, CoT and self-reflection for decision-making. Additionally, error detection and feedback mechanisms are applied during low-level execution.

## 3 EMMOE Benchmark

### 3.1 Problem Statement

EMMOE requires that robots explore environments and perform various open-vocabulary mobile manipulation tasks based solely on language instructions and sensor observations. More specifically, it combines embodied task planning, embodied decision making, visual language navigation and manipulation in continuous space, which requires highly on both level of models and the design of the agent system.

Table 1: **Dataset Comparisons.** EMMOE-100 is the first dataset to integrate mobile manipulation tasks with embodied task planning, decomposing long mobile manipulation trajectories into discrete actions then executed by low-level policies in continuous space.

| Benchmark | Low-level Policy Selection | Task Planning | Manipulation | Navigation | Procedure Annotations | Re-plan | LMM Trainable Format | COT Analysis | Open-ended Instructions | DPO Sub-dataset |
|---|---|---|---|---|---|---|---|---|---|---|
| OVMM | ✗ | ✗ | Continuous | Continuous | ✗ | ✗ | ✗ | ✗ | ✗ | ✗ |
| BEHAVIOR-1K | ✗ | ✓ | Continuous | Continuous | ✗ | ✗ | ✗ | ✗ | ✗ | ✗ |
| ALFRED | ✗ | ✓ | Discrete | Discrete | ✓ | ✗ | ✓ | ✗ | ✗ | ✗ |
| Octopus | ✗ | ✓ | Discrete | Discrete | ✓ | ✓ | ✓ | ✓ | ✗ | ✗ |
| Habitat-Lab 2.0 | ✗ | ✗ | Continuous | Continuous | ✗ | ✗ | ✗ | ✗ | ✗ | ✗ |
| VirtualHome | ✗ | ✓ | Discrete | ✗ | ✓ | ✗ | ✓ | ✗ | ✗ | ✗ |
| ManiSkill-2 | ✗ | ✓ | Continuous | Continuous | ✗ | ✗ | ✗ | ✗ | ✗ | ✗ |
| Grutopia | ✗ | ✓ | Continuous | Continuous | ✗ | ✗ | ✗ | ✗ | ✗ | ✗ |
| **EMMOE-100** | ✓ | ✓ | Continuous | Continuous | ✓ | ✓ | ✓ | ✓ | ✓ | ✓ |

## 3.2 EMMOE-100 Dataset

By controlling Fetch Robots (Fetch Robotics, 2020) in Habitat-Lab 2.0 (Szot et al., 2021), we collect EMMOE-100, a dataset consists of 100 complex everyday tasks. We sample 100 different scenarios from Replica Challenge (Szot et al., 2021) to build simulation environments. In each scene, we'll first design a daily mobile manipulation task, then manually control a Fetch robot to complete the task in continuous space and decompose execution trajectories into discrete subtasks. When designing tasks, we ensure their feasibility in simulation. Therefore, we select six of most common fundamental skills based on Habitat's capabilities and support, which are subsequently composed into our task sequences. Each subtask consists of an executable action, a target, and a low-level model selection, with the total steps constrained to a maximum of 15. Finally we obtain 966 subtasks in total. We also annotate each subtask with four first-person view images and detailed reasoning processes. Moreover, we intentionally design some failed subtasks and provide re-plans to enhance dataset robustness. To alleviate grounding problems, we construct SFT and DPO sub-datasets, which will be introduced in Section 5.1.

To enhance task diversity and better align with human demands, we design tasks with five different attributes: **Short-horizon** tasks like *pick something and place it somewhere*. **Long-horizon** tasks which consist of at least ten subtasks. **Open-ended** tasks that allow multiple results and solutions. **Logical** tasks that provide vague descriptions and require logical reasoning. **Human-style** tasks are described in a natural conversation style. One task can possess multiple attributes as some of these attributes are not contradictory. Table 1 shows detailed comparisons with other mobile manipulation and embodied task datasets. We also provide detailed task statistics in Appendix B.

## 3.3 Evaluation Metrics

The most fundamental metrics in embodied task planning are Success Rate (SR) and Goal-Condition Success (GC) (Shridhar et al., 2020). SR measures the proportion of successful trajectories, while GC is the ratio of goal conditions achieved at the end of a trajectory. A trajectory is considered successful only if GC reaches 100%. However, GC focuses only on the final result and relies on pre-defined state goals, thus failing to meet the requirements of our EMMOE tasks, which require fine-grained and language-based evaluations. Although some studies (Li et al., 2024a) conduct more fine-grained evaluations, they overlook the flexibility and coherence in embodied task planning and still rely on abstract terms. The success of an individual step may not contribute to the final success, and an output that differs from the ground truth but can complete the task in an alternative way should not be considered incorrect. Furthermore, fine-grained evaluation of the entire agent system remains a challenge. To overcome these limitations and provide more diverse evaluations, we propose the following new metrics. All details about definitions and visible calculation examples can be found in Appendix C.

**Task Progress** To better measure the task execution process and the interrelations among subtasks, we propose Task Progress (TP), which is calculated as follows:

$$TP = \max_{k_i \in K_T} \left( \frac{\text{len}(k_i^{\text{check}})}{\text{len}(k_i)} \right) \tag{1}$$

A keypath is defined as an ordered node set of all necessary subtasks required to complete a task, $k_i$ is the $i$-th keypath in the keypath set $K_T$ for task $T$, each task is assigned with several keypaths, representing different ways to complete the task. We strictly match the execution trajectory with the subtask nodes in $k_i$ in sequential order. Once the node in $k_i$ is successfully matched, it will be added to another ordered set $k_i^{\text{check}}$, then the ratio between the length of $k_i^{\text{check}}$ and the length of $k_i$ will be recorded. This process will be repeated for all keypaths in $K_T$, and the highest ratio will become the TP value of the trajectory. Only if TP reaches 100%, the trajectory will be considered successful. TP considers both the flexibility of the execution process and the relationships between every step. The way of using natural language and execution results to evaluate also simplifies new task design and enables evaluation in real-world scenarios, where writing PDDL files is impractical.

**Success End Rate** A fully autonomous robot should be able to actively terminate the execution at a proper moment. Otherwise, even if the task is already done, the robot may continue running and get stuck in an endless loop. Therefore, we propose Success End Rate (SER) to evaluate whether the agent has the ability to understand its current situation and reasonably determine the appropriate timing for task termination, the calculation method is as follows:

$$SER = \frac{\text{len}(S)}{\sum_{t \in M} \text{count}_t(\text{end})} \tag{2}$$

$t$ represents a single trajectory and $M$ is the set of trajectories for all tasks, $\text{count}_t(\text{end})$ equals 1 if $End$ is the final action of $t$ or 0 otherwise. $S$ is the set of successful trajectories, of which TP equals 100%. Then SER is calculated as the ratio of the number of successful trajectories to the number of trajectories that the agent deemed successful. Once SER reaches a certain threshold or even 100%, auxiliary methods or metrics are no longer needed to calculate SR.

**Success Re-plan Rate** Execution failures are common cases in the real world, especially in unfamiliar environments, which makes the ability to quickly adjust from failures and continuously adapt to new environments a crucial skill. To measure the adaptation and generalization abilities of the agent, we propose Success Re-plan Rate (SRR), which is calculated as follows:

$$SRR = \frac{\sum_{t \in S} \text{count}_t(\text{replan})}{\sum_{t \in M} \text{count}_t(\text{replan})} \tag{3}$$

a replan is defined as the next action that agent takes after the previous action failed, $\text{count}_t(\text{replan})$ is the number of re-plans in trajectory $t$, other symbol definitions are the same as SER. SRR represents the effectiveness of re-planning and adaptability of the agent. When SRR reaches 100%, it indicates that the agent can adapt to all failures and then successfully complete the task.

## 4 HomieBot

### 4.1 Overview

In this section, we will introduce how HomieBot accomplishes EMMOE tasks. HomieBot employs a hierarchical framework with communication mechanisms for interleaved execution. High-Level Planning (HLP) deals with embodied decision making and planning adaptation, while Low-Level Execution (LLE) translates subtasks into continuous low-level controls and provides feedback to HLP. We will describe HLP in Section 4.2 and LLE in Section 4.3. A system overview is shown in Fig.2.

### 4.2 High Level Planning (HLP)

A long trajectory will be decomposed into several subtasks, the agent must continuously interact with the environment and adjust plans based on real-time feedback to ensure generated subtasks are practical. We design elaborate input and output instructions to facilitate dynamic adjustments during execution. Video-LLaVA (Lin et al., 2023) is selected as our planner model $M$ and fine-tuned with SFT and DPO sub-datasets, which will be described in Section 5.1.

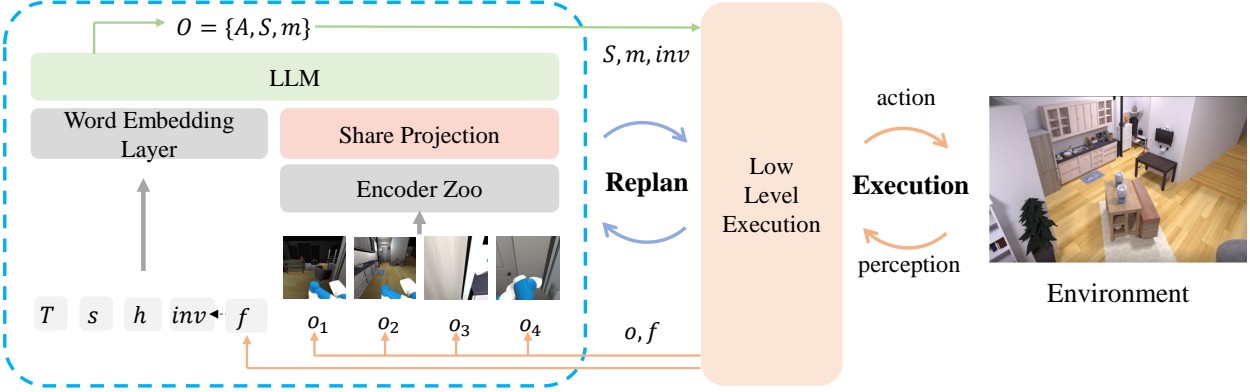

Figure 2: **Overview of HomieBot.** HomieBot leverages a hierarchical framework to handle long-horizon tasks: High-Level Planning decomposes tasks into manageable actions, Low-Level Execution accomplishes received actions and provides real-time feedback.

**Multi-modal Instruction**    To help the LMM better understand current situations, the format of the input instruction $I$ is as follows:

$$I = \{o_{1\sim4}, s, T, inv, h, f\} \tag{4}$$

In the visual component, four first-person view images $o_{1\sim4}$ correspond to four directions respectively. In the textual component, system information $s$ and user task $T$ remain constant throughout the conversation, reminding the agent of its responsibility. Feedback $f$ indicates the status of the last execution and detailed error information if failed, it will also be used to update other parts in $I$. Inventory $inv$ reflects the items currently held by the agent, primarily to prevent the generation of illogical actions, it is updated based on both $f$ and the type of the last action. Execution history $h$ logs all previous subtasks and their results. Once receiving $f$, the last subtask and its result will be logged in $h$. Besides, to better align with real-world scenarios, we prohibit directly inputting background information into the LMM (e.g. raw object data, Bird's Eye View images etc.). The planner must explore the environment and enhance its intrinsic capabilities to generate more reasonable outputs.

**Json-format Output**    Considering that different low-level policies may require different information formats and to facilitate the replacement and maintenance of each module, we define our output in the following uniform format:

$$O = M(I) = \{A, S, m\}, S = \{\texttt{action}, \texttt{target}\} \tag{5}$$

$A$ represents the analysis of each output, which is inspired by works like CoT (Wei et al., 2022). Before generating final outputs, planner model $M$ is expected to summarize previous executions and current situations, analyze what to do next, and propose the subsequent subtask $S$. To ensure the feasibility of the output, `action` can only be chosen from the available action list. Similarly, $m$ which represents the selected low-level models or policies, is also restricted to a given model list. `target` can be either an object or a spot, it should be observable in the provided images and deemed necessary to complete the task.

### 4.3    Low Level Execution (LLE)

LLE will convert $S$, $m$ and $inv$ from HLP into precise model-calling instructions. Error detection will be applied at different stages to monitor the whole process. Once the execution is completed or failed, environmental images and feedback will be sent back to HLP. We set up six skills based on the support of the simulator(see Table E1). Since the required information varies from models and would significantly impact the model performance, we establish two distinct settings to ensure fairness.

**Execution With Background Information**    More specifically, execution with background information means that the selected model needs precise position and state information of the target. As M3 (Gu et al.,

2022) shows exceptional performance in all skills when utilizing background information in Habitat, we choose it as the unique model choice $m$ in this setting. To make M3 adapt to our task requirements, we implement a name mapping for *target* and adjust its original setting to better align with requirements of our tasks. In addition to text and image data, LLE also captures the execution process of each step and the entire trajectory data in video format. This means that HomieBot has the potential to bridge the gap between robot data and LMM data as the entire execution process is fully automated and annotated, users only need to set up the scene and input instructions. The video data can be utilized for IL in robotics, while the text and image data can be utilized for LMM training.

**Execution Without Background Information**   Without background information means that the agent can only rely on the information captured by its sensors and the intrinsic abilities of low-level models to complete the task. As shown in Table E2, we set two manipulation models and two navigation models. For manipulation, RT-1-X (Padalkar et al., 2023) is used for *Pick* and *Place*, while Octo Team et al. (2024b) is set for *Open* and *Close*. For navigation, NoMaD (Sridhar et al., 2024) specializes in image navigation and is suitable when *target* is a spot or large object. PixNav (Cai et al., 2024) excels in pixel-level and object navigation, making it ideal when *target* is a detectable object. As the deployment of robots in the real world always demands high real-time performance and is constrained by hardware limitations, we prefer to choose lightweight models rather than the currently popular VLA models to prevent the system from becoming too burdensome. Additionally, breaking down long-horizon tasks into action primitives would also reduce the performance requirements of low-level models. Compared to general-purpose end-to-end models, specialized lightweight models can complete the action while reducing time costs.

**Error Detection**   To facilitate communication with HLP and provide more detailed error information, we further classify common errors into four main types and several subtypes. **Logical error** *L1*: The agent's hands are already full but still attempts to pick/open/close; *L2*: The agent holds nothing but attempts to put; *L3*: The agent attempts to pick/put the object in a closed container; *L4*: The agent attempts to interact with a non-interactive object. **Distance error** *D1*: The agent stands too far and is unable to reach the target; *D2*: The agent is too close to the target and its arm is hindered from properly extending during interaction. **Format Error** *F1*: The output action or model is not in the available list; *F2*: The output target does not exist in the current scene or can not be recognized by low-level models. **Execution Error** *E1*: The limited capabilities of the low-level models or policies cause the failure; *E2*: Failed execution may result in the inventory information being accidentally updated. Furthermore, we conduct multiple phases of error detection during the whole process to guarantee the executions. More classification and detection details are given in Appendix E.

## 5   Experiments

### 5.1   Data Augmentation

**SFT Augmentation**   Previous work (Zhang et al., 2024c) has shown that a standardized data format would significantly enhance model training and evaluation. Therefore, we write a uniform script to convert the original EMMOE-100 data into fixed-format conversation data. During this process, all failed subtasks will be skipped as they are treated as junk data for the SFT dataset, and we initially obtained 930 SFT data in this way, which is still insufficient for LLM training. To expand the dataset, we use GPT-4o (Hurst et al., 2024) to regenerate text descriptions of tasks and the analysis of each subtask for three times. This approach not only enhances the diversity of instructions, allowing the LLM to adapt to different user input styles, but also helps to avoid introducing additional inaccuracy or inconsistency. Finally, we obtain 3,720 SFT data in total. The relevant code and data samples are available in Appendix F.1.

**DPO Augmentation**   DPO (Rafailov et al., 2024) training has a strict requirement for data format, which must include *prompt*, *chosen* and *rejected*. For the $i$-th subtask and its input instruction $I_i$, if the execution of output $O_i$ fails but the next output $O_{i+1}$ succeeds after re-plan, we will choose $I_i$ as the *prompt*, $O_i$ as the *rejected* and $O_{i+1}$ as the *chosen*. Although this approach aligns well with the concept of preference data, the proportion of re-planned data is relatively low. Thus, we utilize following methods to construct new DPO

data. **Order Change**: We shuffle the order of successful subtasks, treating successful output $O_i$ as *chosen* and $O_{i+1}$ as *rejected*. This approach aims to help LLMs learn the logical relationships between subtasks, particularly the proper sequence of actions. **Action Change**: To standardize the planner model's output and reduce responses outside the action list, we replace actions in subtasks with non-standard names or actions outside the available list. **Model Change**: To enable the LLM to own the ability to select the appropriate low-level model for a given scenario, we replace the model choice with models of the same type in the model list. As a result, we get 10,104 DPO data in total. More processing flows and data samples are provided in Appendix F.2.

## 5.2 Model Training

We select 90 tasks from EMMOE-100 as our training tasks. Using the methods described in Section 5.1, we obtain 3,316 SFT training data and 8,984 DPO training data in total. Then we select Video-LLaVA-7B (Lin et al., 2023) as our base model and conduct a two-stage training process. In the first stage, we fine-tune the base model with a learning rate of 5e-4 on 4×NVIDIA A40. In the second stage, we align the fine-tuned model with DPO and train with a learning rate of 5e-6. To prevent catastrophic forgetting and maintain the intrinsic model capability, LoRA (Hu et al., 2021) is applied in both stages, with LoRA rank set to 128 and $\alpha$ to 256 in stage one, and LoRA rank set to 8 and $\alpha$ to 8 in stage two.

## 5.3 Setup

**Metrics**  In addition to SR, TP, SER and SRR introduced in Section 3.3, we also choose Path Length Weighted SR (PLWSR) (Shridhar et al., 2020) as one of our evaluation metrics. PLWSR is defined as SR×(length of successful trajectory) / $max$(length of expert trajectory, length of successful trajectory) and measures the ability gap between the agent and the expert in successful trajectories.

**High Level Planner**  In baseline planner selections, GPT-4o (Hurst et al., 2024) and Gemini-1.5-Pro (Team et al., 2024a) are the most popular and common closed-source models. The reasoning model o1 (Jaech et al., 2024) is famous for its powerful reasoning abilities and can effectively handle complex inference tasks. For open-source models, as the base planner model - VideoLLaVA can hardly finish EMMOE tasks, making the evaluation meaningless, we instead choose Qwen2-VL-7B (Wang et al., 2024b) and MiniCPM-V 2.6 (Yao et al., 2024), which perform well in various multi-modal tasks and have similar model sizes. GPT-4o, Gemini-1.5-Pro and o1 can be easily integrated into HomieBot after minor adjustments to format requirements. By leveraging the in-context learning abilities and providing output examples for each inference, the other two models can also be deployed in our system.

**Low Level Executor**  As the model without finetuning performs poorly due to the impact of the real-to-sim gap, we currently focus on evaluating the individual skills in M3 (Gu et al., 2022). We extract and modify implementations of each skill. Original skills require the initial and final states of the object. We map the object name to obtain specific background information and select the nearest object. Additionally, robotic arms will be reset after each execution to enhance the success rate. We also pass all environmental state information between executions to ensure environmental consistency. We use single NVIDIA A40 to run both models and provide more details in Appendix H.1.

**Evaluation Benchmark**  All tasks in EMMOE-100 will be used for evaluation, and the remaining ten untrained tasks will serve as our test set. Each task is executed three times with a maximum step limit of 20 each time, the average execution results will be used for the final calculation.

## 5.4 Results

We begin with a general evaluation since all data are unseen to baseline models. As shown in Table 2, the DPO version of HomieBot achieves the best performance in SR, PLWSR, TP and SER metrics. o1 also demonstrates excellent performance in our tasks and surpasses the SFT version in some metrics. Additionally, it is evident that for open-source models of similar size, even state-of-the-art LMMs like Qwen2-VL-7B (Wang

Table 2: Performance comparison of different models on EMMOE-100 tasks. The highest values for each metric are highlighted in **bold**.

| Model | SR | PLWSR | TP | SRR | SER |
|---|---|---|---|---|---|
| QWEN2-VL-7B (WANG ET AL., 2024B) | 1.00 | 0.50 | 16.55 | 0.59 | 25.00 |
| MINICPM-V 2.6 (YAO ET AL., 2024) | 0.67 | 0.57 | 14.45 | 0.06 | 40.00 |
| GPT-4O (HURST ET AL., 2024) | 13.33 | 10.51 | 29.79 | 3.57 | 49.38 |
| GEMINI-1.5-PRO (TEAM ET AL., 2024A) | 17.33 | 14.79 | 38.03 | 3.39 | 55.91 |
| O1 (JAECH ET AL., 2024) | 28.67 | 24.11 | 44.52 | **13.80** | 38.57 |
| HOMIEBOT-7B (SFT) | 27.67 | 20.88 | 50.27 | 9.23 | 53.90 |
| HOMIEBOT-7B (SFT+DPO) | **30.30** | **24.66** | **51.39** | 8.72 | **60.81** |

Table 3: Performance comparison of HomieBot on the training and test split. The highest values for each metric are highlighted in **bold**.

| Model | Train split | | | | | Test split | | | | |
|---|---|---|---|---|---|---|---|---|---|---|
| | SR | PLWSR | TP | SRR | SER | SR | PLWSR | TP | SRR | SER |
| HOMIEBOT (SFT) | 28.52 | 21.49 | 50.16 | 9.59 | 53.85 | **20.00** | **15.36** | **51.19** | **6.55** | 54.55 |
| HOMIEBOT (SFT+DPO) | **31.84** | **25.82** | **52.29** | **9.69** | **60.71** | 16.67 | 14.36 | 43.39 | 3.08 | **62.50** |

et al., 2024b) and MiniCPM-V 2.6 (Yao et al., 2024) struggle in EMMOE tasks without additional training. The low overall success rate is primarily due to the difficulty of EMMOE tasks, which reflects that there is still a long way from achieving truly autonomous home robots. Moreover, compared with GPT and Gemini which have much larger parameter sizes, HomieBot still achieves improvements, and shows substantial gains over models of a similar scale. This validates the effectiveness of our dataset and augmentation methods.

For SER, though the DPO version still performs best, the improvement is not so obvious. This phenomenon should be attributed to the nature of SER, which reflects the model's ability to correctly determine when a task is completed and should be terminated. It is less influenced by format requirements and low-level executions, but relies more on the model's inherent reasoning ability. The strong reasoning capabilities of GPT-4o (Hurst et al., 2024), Gemini-1.5-Pro (Team et al., 2024a) and o1 (Jaech et al., 2024) enable them to effectively decide when to end a trajectory. Moreover, as we observed during experiments, o1 is more likely to "give up" compared with other models. After several failed attempts, o1 would judge that the current task is infeasible and directly terminate task execution, while other models would continue to explore. This tendency results in relatively lower SER, and also affects TP to some extent.

For SRR, o1 performs best and SFT version performs better than DPO version. Since SRR reflects the model's ability to adapt to environments and adjust from failure, this result suggests that o1 can better leverage feedback information to make more effective re-plans. Besides, it could also be relevant with the limitations of the DPO method (Xu et al., 2024). Although DPO brings unparalleled advantages in training efficiency and convenience, it compromises the model's generalization and transferability to certain extent. Therefore, we further evaluate HomieBot separately in training and test set. As we can observe in Table 3, while DPO version performs best on all metrics in the training split, it only outperforms SFT version on SER in the test split. Additionally, DPO version shows a significant decline on SRR. This observation further confirms that the DPO method introduces certain generalization issues.

Notably, SER remains stable for both versions across the training and test splits, which further demonstrates that SER is more related to the model's inherent judgment ability, and our specialized handling of *End* during dataset construction has enhanced this ability (See in Appendix F.2).

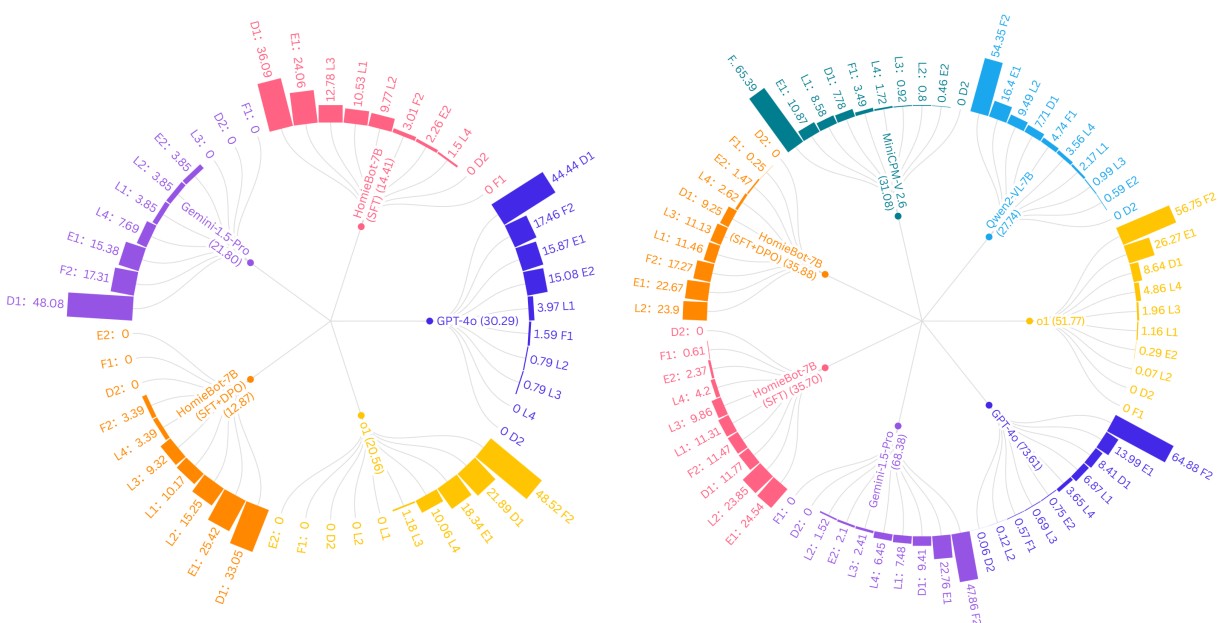

Figure 3: **Error Statistics.** The left and right figures depict the proportion of each error type of each model in successful and failed trajectories respectively. Additionally, we indicate the proportion of total execution failures next to each model's name. Due to too few successful trajectories for Qwen2-VL and MiniCPM-V 2.6, their results will not be shown in the left figure. The original statistical data are available in Appendix H.2.

## 5.5 Analysis

To further explore the reasons for the overall low success rate and demonstrate how HomieBot can be used to simultaneously evaluate both HLP and LLE, we conduct a detailed analysis based on the results in Section 5.4. Using the error classification in Section 4.3 and the recorded feedback, we collect all errors that occurred during experiments. To identify which errors are acceptable and solvable and which are the primary causes of failure, we further classify the collected errors according to whether they appear in successful or failed trajectories, the results are shown in Figure 3.

**Error Analysis** Except for $E1$ and $E2$ error that come from low-level models, each error type corresponds to different capabilities of LMMs. In failed trajectories, the predominant error type across all baseline models is $F2$ error. This suggests that the primary obstructive factors are physical grounding failures and model hallucinations. In practical execution processes, we observe that even models are already told the object doesn't exist or can't be recognized, they may still produce inappropriate outputs or repeat mistakes after several steps. This issue has been significantly improved in our models, which also highlights the significance of LMM-trainable format data. With a small amount of augmented data, LMM can build up a general understanding of current environments, enabling outputs to be compatible with low-level models.

Besides, the proportion of failed executions for two open-source models is relatively low, indicating that most subtasks are successfully completed, which seems to conflict with the very poor SR. Based on our observations, since EMMOE includes numerous complex and long-horizon tasks, execution histories often become lengthy. When the model's understanding ability is insufficient, it may fail to fully understand or even forget previous execution contents, ultimately resulting in meaningless outputs. Although these subtasks can be successfully executed, they contribute nothing to the final task, and even worse, they will consume remaining steps and fasten task termination. In successful trajectories, the most common error is $D1$ error. This indicates that even when the model's spatial perception ability is insufficient, it can be adjusted through

Table 4: Results of LLE evaluations. P represents the proportion of single action errors. SR here represents an average value as each skill is attempted up to three times per execution.

| Metrics | Go to | Pick | Place | Open | Close |
|---------|-------|------|-------|------|-------|
| P | 38.49 | 49.77 | 7.30 | 3.32 | 1.11 |
| SR | 45.32 | 22.45 | 40.97 | 43.13 | 36.45 |

Table 5: The performance of each type of task is presented in the format SR (PLWSR).

| Model | Short-horizon | Long-horizon | Open-ended | Logical | Human-style |
|-------|---------------|--------------|------------|---------|-------------|
| HOMIEBOT-7B (SFT) | 43.75 (32.31) | 24.60 (18.70) | 18.52 (11.93) | 34.01 (25.45) | 25.24 (18.70) |
| HOMIEBOT-7B (SFT+DPO) | 41.67 (34.24) | 28.11 (22.82) | 15.38 (11.57) | 35.86 (28.05) | 27.88 (21.78) |

feedback information. Typically, after a $D1$ error occurs, the model will output *Go to* action based on the feedback, effectively resolving this error.

**Case Study**  Based on the trajectories from the experiments, we further analyze the execution performance and failure factors, then obtain the following preliminary conclusions: **Terrible Grounding.** Even after informing the model that the target doesn't exist, the model may still generate incorrect outputs, or forget the mistake after a few steps. **Limited LLE.** Though high-level planner makes correct plans, execution continues to fail due to the limited ability of low-level models. **Meaningless Outputs.** All outputs are successfully completed, but the agent keeps circling in place without making progress. These meaningless outputs quickly consume the remaining execution steps, ultimately causing the task to fail. **Solvable D1 Error.** After a $D1$ error happens, a *Go to* action can effectively solve it and facilitate the success of the whole trajectory. We provide corresponding cases in Appendix I.

**LLE Evaluation**  Comprehensive error types allow us to evaluate HLP and LLE separately. We further classify $E1$ and $E2$ errors based on action types and count total occurrences of each action, the calculation results are shown in Table 4. It is evident that *Pick* action has a significantly lower success rate and the highest proportion of execution errors compared to other actions.

**Task Performance**  We also evaluate SR and PLWSR for each type of task defined in Section 3.2. As shown in Table 5, short horizon tasks are relatively easy due to straightforward processes and fewer overall steps. The most challenging are open-ended tasks, which usually have a very long total step count, with flexible processes and results, demanding powerful capabilities from both HLP and LLE models.

## 6  Discussions

**Conclusions**  We propose EMMOE, the first unified benchmark designed to evaluate both high-level planners and low-level policies. Then we present the collection and features of EMMOE-100 and propose three novel metrics to complement existing evaluation methods. Next, we introduce our HomieBot and illustrate how its two main components HLP and LLE function. In experiments, we demonstrate how to construct LMM-trainable SFT and DPO datasets and evaluate different levels of models. Finally, we conduct an in-depth analysis based on the detailed error information.

**Limitations and Future Work**  Limited actions and available space in Habitat restrict the scope of task design. Besides, standardized output will sacrifice certain information precision. The growing number of model inferences will also lead to additional time costs. Moreover, we conduct deployment and experiments only in simulation. This decision was driven by two key factors: 1) All researchers are required to make assessments under the same conditions, thus ensuring optimal fairness and consistency. 2) Since sufficient GPU resources for deploying LLMs and a capable mobile robot with manipulators are unaffordable for most

researchers, conducting evaluations on a simulation platform could effectively lower the research barrier. In the future, we'll collect more tasks, design a more efficient system, and explore real-world evaluations.

**Sim-to-real Gap**    Since the simulator is not exactly the same as the real world, some unrealistic phenomena will occur during task execution. For example, the robot arm may shake drastically when a collision occurs, or the robot may directly enter the interior of the object. Besides, real world environments could not be fully realized in simulation, and the physical properties of objects in the simulation also differ from those in the real world. As a result, some operations that are feasible in the real world cannot be executed in simulation, policies trained in simulation may not generalize well to more complex scenarios in the real world. Finally, the design of the Fetch robot also affects task execution. Since its manipulator cannot reach the ground, if an object is accidentally dropped during the picking or moving process, it can no longer be picked up again. This is also one of the reasons why the success rate of *Pick* is so low.

## Broader Impact Statement

This research utilizes publicly accessible models and simulators, ensuring that all data is anonymized and compliant with privacy regulations. However, we recognize the ethical implications of deploying such agents in human environments. Physical safety remains the primary concern; current low success rates highlight significant risks, such as object drops or collisions. As all experiments were conducted in simulation, any future real-world deployment will require strict hardware safeguards, compliant control mechanisms, and human supervision. We also acknowledge the limitations of our experimental setting: relying solely on the Replica environment may not fully represent the diversity of real-world household settings. Furthermore, the socioeconomic impact of household automation is substantial. While our goal is to enhance accessibility and alleviate domestic burdens, we advocate for responsible deployment aligned with social equity and fair labor transitions. Finally, regarding privacy, future applications must employ on-device processing and consent-based data governance to prevent misuse. To support accountability and reproducibility, all code and models will be openly shared.

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

# Appendix

The appendix is structured as follows:

## A    Related Work

### A.1    Embodied Tasks and Benchmarks

As embodied agents and LLMs develop rapidly, many embodied tasks and benchmarks have emerged. In Embodied Question Answering (EQA) tasks, EQA-v1 (Das et al., 2018), VirtualHome (Puig et al., 2018), MT-EQA (Yu et al., 2019), MP3D-EQA (Wijmans et al., 2019), IQUAD V1 (Gordon et al., 2018), OpenEQA (Majumdar et al., 2024), HM-EQA (Ren et al., 2024), S-EQA (Dorbala et al., 2024) contains a variety of task range to evaluate logical reasoning abilities of LLMs. BLINK (Fu et al., 2024a) for visual perception abilities of LMMs. In Vision-and-Language Navigation (VLN) tasks, R2R (Anderson et al., 2018), R4R (Jain et al., 2019) and VLN-CE (Krantz et al., 2020), SOON (Zhu et al., 2021) evaluate LLM's capabilities under different navigation settings. ALFRED (Shridhar et al., 2020) Behavior series (Srivastava et al., 2022; Li et al., 2023a) focus on interactive household tasks OVMM (Yenamandra et al., 2023) involves picking and placing any object in unseen environments. VLA-3D (Zhang et al., 2024b) for 3D semantic scene understanding and navigation. Common manipulation datasets include MT-Opt (Kalashnikov et al., 2021), VIMA (Jiang et al., 2022), ManiSkill2 (Gu et al., 2023), Calvin (Mees et al., 2022), BridgeData-v2 (Walke et al., 2023), RH20T (Fang et al., 2023a), Open-X (O'Neill et al., 2024), AgiBot World (contributors, 2024). In mobile manipulation, RT series (Brohan et al., 2022; Zitkovich et al., 2023) and Mobile ALOHA (Fu et al., 2024b) exhibit strong capabilities. GRUTOPIA (Wang et al., 2024a) takes human participation into account. Additionally, some benchmarks focus on cross-embodiments, like RoboMIND (Wu et al., 2024). EmbodiedBench (Yang et al., 2025) try to evaluate LMMs together in high-levels and low-levels. Despite numerous benchmarks, a unified benchmark and relevant task is still missing. Traditional mobile manipulation uses IL to learn entire trajectories, complicating the evaluation of intermediate processes. In our work, we propose EMMOE, a holistic benchmark designed to assess both final results and the execution process.

### A.2    LLMs For Robotics

LLM-driven embodied agents represent cutting-edge advancements in robotics. SayCan (Ahn et al., 2022), Palm-E (Driess et al., 2023), LLM-Planner (Song et al., 2023) and EmbodiedGPT (Mu et al., 2024) combine LLMs with complex embodied tasks. TAPA (Wu et al., 2023) and SayPlan (Rana et al., 2023) use visual modules for multi-room settings. Voyager (Wang et al., 2023), STEVE (Zhao et al., 2023b), Smallville (Park et al., 2023) and Octopus (Yang et al., 2023a) use LLMs to choose pre-defined functions. L3MVN (Yu et al., 2023), ESC (Zhou et al., 2023), SayNav (Rajvanshi et al., 2023) and VLFM(Yokoyama et al., 2024) build frontier or semantic maps to navigate. ViNT (Shah et al., 2023) and NoMaD (Sridhar et al., 2024) focus on image navigation, PixNav (Cai et al., 2024) uses LLM to select target image pixel. GOAT (Chang et al., 2023)

is a comprehensive navigation system. Navid (Zhang et al., 2024e) and Uni-Navid (Zhang et al., 2024d) focus on end-to-end navigation models. RT-2 (Zitkovich et al., 2023) is the first Visual Language Action (VLA) model. RoboFlamingo (Li et al., 2023b) and OpenVLA (Kim et al., 2024) are open-source VLA models. Leo (Huang et al., 2024a) focuses on multiple QA problems. Manipulate Anything (Duan et al., 2024) and Octo (Team et al., 2024b) are light models for arm control. ALOHA (Zhao et al., 2023a) improves action prediction through action chunking. RoboAgent (Bharadhwaj et al., 2024) enhances object detection and generalization, and LCB (Shentu et al., 2024) uses LLMs to generate implicit strategy goals. ManipLLM (Li et al., 2024b), VoxPoser (Huang et al., 2023), Rekep (Huang et al., 2024b) combine environmental perception and task execution.

### A.3  LLMs for Task Planning

Typical usages of LLM for task planning include treating LLM as a translator or a planner. There are also some studies combining LLMs with traditional PDDL (Guan et al., 2023; Valmeekam et al., 2024; Silver et al., 2024; Zhou et al., 2024), in which LLM will be treated as a translator between the real-world and specific domain planner. But this method is limited by the performance of the domain planner and can't leverage the powerful commonsense reasoning capabilities of LLMs to assist in planning. When LLM is treated as a planner, discrepancies between LLM's outputs and real-world conditions always lead to execution failures. LLM-Planner (Song et al., 2023) make a straightforward re-plan. Self-Refine (Madaan et al., 2024) use single LLM as generator and evaluator. Reflexion (Shinn et al., 2024) treats LLM as the Actor and the evaluator as the Critic. ViLA (Lin et al., 2024) utilizes GPT-4V (Yang et al., 2023b) to obtain visual feedback. However, self-improvement relies heavily on prompt design and intrinsic capabilities of LLMs. If errors unrelated to planning occur, LLMs may struggle to self-correct. Inner Monologue (Huang et al., 2022) and RoCo (Mandi et al., 2024) utilizes external collision detection and feedback. DoReMi (Xie et al., 2024) sets pre-defined constrains. Nevertheless, LLMs may make same mistakes in similar situations as the model weights are not changed. SayCan (Ahn et al., 2022) trains a value function to consider both generated actions and their values. Remember (Zhang et al., 2024a) builds a memory module and retrieves similar state-action pairs. Retroformer (Yao et al., 2023) learns a retrospective model via policy gradient optimization. While RL-based adaptation mechanisms are able to adjust actions before execution, defining and training an effective value function or reward model is highly challenging. The recently popular DPO (Rafailov et al., 2024) algorithm greatly simplifies this process by requiring only a preference dataset. In our HOMIEBOT, we use DPO for model alignment, CoT (Wei et al., 2022) and self-reflection for decision-making. Additionally, error detection and feedback mechanisms are applied during low-level execution.

## B  Dataset

### B.1  Data Collection

We first randomly sample episode information provided by Replica Challenge (Szot et al., 2021) to build the task scenario, then we use the Fetch robot to collect EMMOE-100 in Habitat-lab v0.2.3. To facilitate data collection, we modify the original interaction script, and new interface can be seen in Fig. B1. The interface provides both third-person and first-person view observation to facilitate data collection, third-person observation is used to assist with collection, only first-person observation with $256*256$ resolution will be saved. Notably, we only use the scene information to collect environment data, other functions and metrics in Replica Challenge are irrelevant to our work.

We also show the modified code clip, once a single subtask is finished, we can conveniently save relevant information by pressing the keyboard.

```
def save_first_view_images():
    directions = ['left', 'back', 'right', 'front']
    global h_cnt

    h_cnt += 1
    for i in range(4):
        for j in range(19):
```

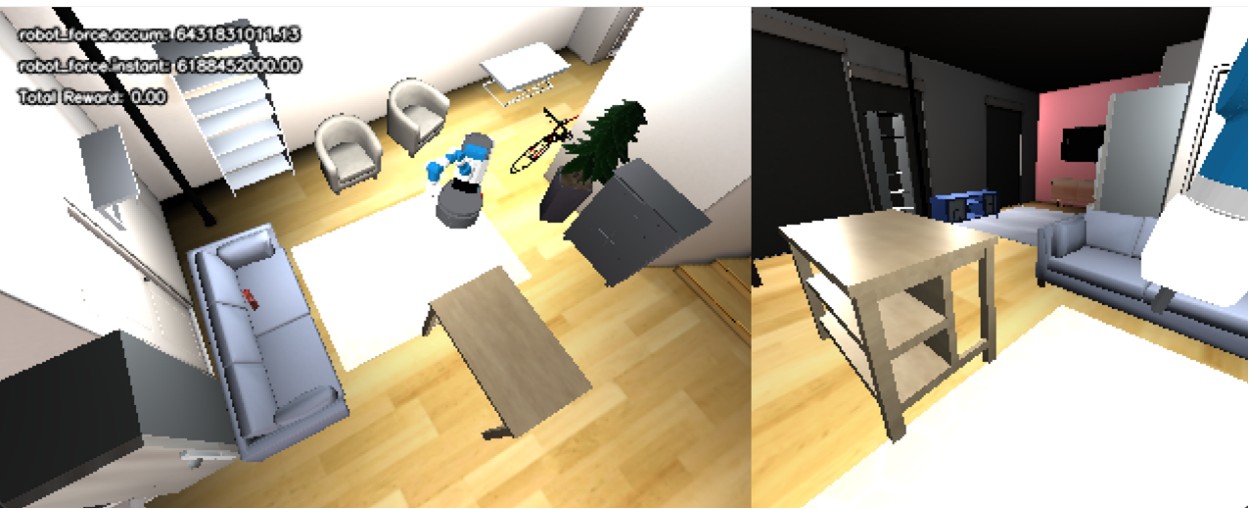

Figure B1: **Data collection interface in Habitat-lab v0.2.3.** Third-person observation in the left is used to facilitate data collection, only first-person observation with 256∗256 resolution in the right will be saved.

```
        base_action = [0, 1]
        name = base_action_name
        args = {base_key: base_action}
        result = step_env(env, name, args)

    use_ob = observations_to_image(result, {})
    draw_ob = use_ob[:]
    from PIL import Image
    ob = Image.fromarray(draw_ob)
    width, height = ob.size
    ob.crop((384, 0, width, height)).save(os.path.join(info_folder, f"
    subtask{h_cnt}_{directions[i]}.png"))

return result, arm_action, end_ep
```

## B.2 Dataset Details

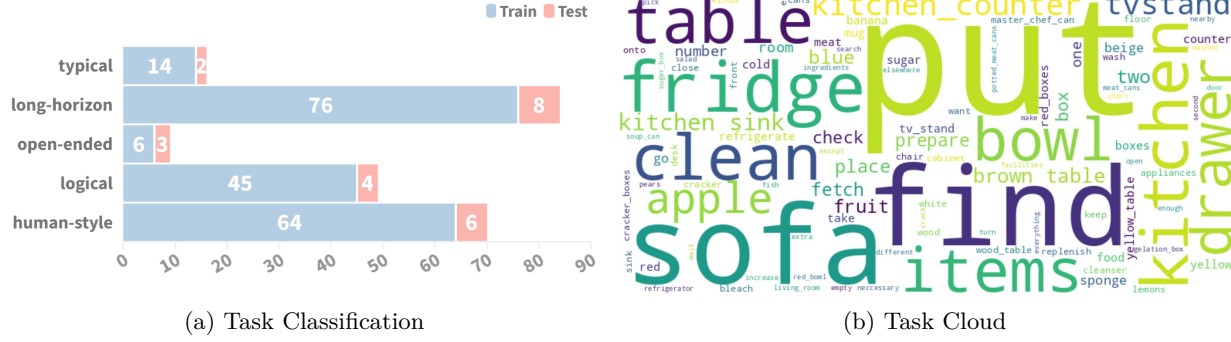

(a) Task Classification        (b) Task Cloud

Figure B2: Dataset Statistics

In terms of task classification, the long-horizon task is the most, with 84 (76 in the train set and 8 in the test set). The least task is the most difficult open-ended task, with 6 in the train set and 3 in the test set. The

distribution of the five types of tasks in the train and test sets is also approximately the same. In the word cloud map, we can see that put, find, sofa, etc. are popular words in our task.

**B.3 Dataset Demonstration**

Here we provide a demonstration of EMMOE data, its viusal information is shown in Fig. B3, the collection method is as Section 3.2.

```
Task: check if there are anything in the kitchen drawer, if it's empty put a
short can into it

Subtask1: [Go to, blue_sofa] (PixNav)
Analysis: According to my final task, I must go to kitchen first, but I can't
find kitchen in my sight and I find a blue sofa, so I should go to the blue
sofa first and find kitchen later
Feedback: None
Inventory: None

Subtask2: [Go to, living_room] (NoMaD)
Analysis: I successfully arrived sofa but I still can't find kitchen, as I see
 living room, I should go to living room then keep searching kitchen
Feedback: None
Inventory: None

Subtask3: [Go to, kitchen_drawer] (NoMaD)
Analysis: I successfully arrived the living room and find the kitchen drawer,
so I should go to kitchen drawer next
Feedback: None
Inventory: None

Subtask4: [Open, kitchen_drawer] (octo)
Analysis: I successfully arrived the kitchen drawer but it's closed. To check
items in the drawer, I should open it first
Feedback: None
Inventory: None

Subtask5: [Go to, kitchen_counter] (NoMaD)
Analysis: I successfully opened the drawer and find nothing in it, according
to my task, I should put a short can into it in this case, as I find one on
the kitchen counter, I should go there next
Feedback: None
Inventory: None

Subtask6: [Pick, short_can] (RT-1-X)
Analysis: I successfully arrived the kitchen counter and find the can, so I
should pick it next
Feedback: None
Inventory: short_can

Subtask7: [Go to, kitchen_drawer] (NoMaD)
Analysis: I successfully picked the short can and I should come back to
kitchen drawer to put it in
Feedback: None
Inventory: short_can

Subtask8: [Put, short_can, kitchen_drawer] (RT-1-X)
Analysis: I successfully arrived the kitchen drawer and I should put the can
into it next
Feedback: None
```

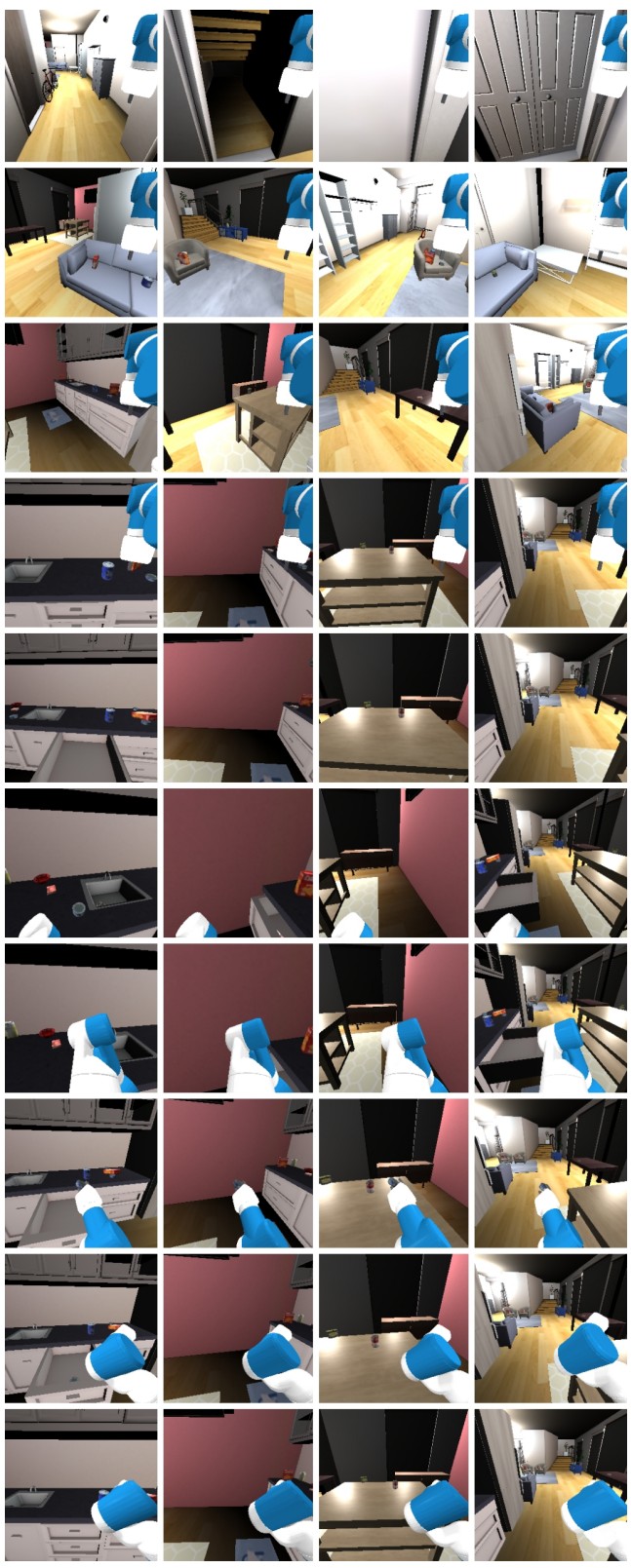

Figure B3: **Task Demonstration.** line: subtask sequence; row: front, left, back, right

```
Inventory: None

Subtask9: [Close, kitchen_drawer] (octo)
Analysis: I successfully put the can into the drawer, and it's better to close
 the drawer next
Feedback: None
Inventory: None

Subtask10: [End]
Analysis: According to the historical execution and final task, I have finally
 finished the task and it's time to end
Feedback: None
Inventory: None
```

We also provide all designed tasks here, the task design principles focus on reflecting human's real-life with a variety of common demands and task descriptions.

```
(1) fetch a frozen meat can and put it on the kitchen counter
(2) clean up the brown table and place all items in the fridge
(3) find a cold apple and put it on the kitchen_counter
(4) find an bowl and put it on the sofa
(5) find an master_chef_can on the wood_table and put it into the drawer
(6) go to the floor 2
(7) prepare neccessary ingredients to make a fruit salad and put them on the
yellow_table
(8) keep the number of red_boxes on the yellow_table to 5
(9) search a blue can for me
(10) fetch one crack box and one sugar box and put them on the beige table
(11) find two cracker boxes in the room and put them on the kitchen counter
(12) check if there are apples in the fridge and put one into it if not
(13) pick all fruit on the brown table and put them on the sofa
(14) put the bowl into the kitchen cabinet
(15) find a bleach cleanser and a sponge then place them on the brown table
(16) fetch two apples from the kitchen counter and put them into the fridge
(17) clean the wood table and put all items except mug to the sofa
(18) I want to eat at the brown table and prepare a fish can for me
(19) fetch two cracker_boxes from the kitchen sink and refrigerate them
(20) check and close all kitchen facilities
(21) prepare two bowls on the brown table
(22) fetch two meat_cans from the kitchen and put them on the beige table
(23) find a mug and put it on the tvstand
(24) go to kitchen then put the red box into the drawer and put the red can
into the fridge
(25) find an apple and place it on the tv_stand
(26) clean the tvstand and put all items to the sofa
(27) clean up the tv_stand and put all items in the kitchen drawer
(28) put the sponge and bleach cleanser on the sofa into the kitchen drawer
(29) freeze a sugar_box
(30) put the blue can on the kitchen_counter to the fridge
(31) find two potted_meat_cans and put them on the sofa
(32) clean up the blue table and put all items to the white cabinet
(33) find an apple and put it on the sofa
(34) take a bowl and a meat can from the kitchen and put them on the brown
table
(35) clean up the kitchen sink and put fruit to the fridge other items to the
kitchen_counter
(36) replenish the number of blue cans on the table to 3
(37) find two bowls in the room and put them in the kitchen sink
(38) put all cracker_boxes on the tvstand to the sofa
```

```
(39) take a yellow box and put it into the fridge.
(40) put the apple on the blue table to the sofa
(41) fetch 3 different kinds of fruit and put them on the beige table
(42) I want to eat at the brown table and prepare some fruit for me
(43) put the frozen sponge into the kitchen drawer
(44) put all bowls on the sofa to the kitchen sink
(45) get a can in the fridge and put it on the table
(46) prepare a washed apple then put it on the yellow table
(47) clean up the tvstand
(48) clean up the chair
(49) put everything in the kitchen sink onto the kitchen_counter
(50) wash the bowl on the kitchen_counter
(51) fetch two sugar boxes in the fridge and put them on the brown table, if
there aren't enough sugar boxes in the fridge, find them elsewhere in the room
(52) Prepare a soup_can and a red_bowl on the kitchen_counter
(53) put all the fruit on the kitchen_counter into the sink
(54) put the bowl on the wood_table and the apple on the kitchen_counter to
the kitchen sink
(55) refrigerate all master_chef_cans on the tvstand
(56) clean up the blue sofa
(57) find a gelation_box and put it in the drawer
(58) put the cracker box in the kitchen sink to the sofa
(59) check if there is food on the sofa then put them in the fridge if so
(60) refrigerate all lemons in the kitchen drawer
(61) put all food on the sofa into the drawer
(62) take the bowl on the table to the kitchen
(63) clean up the tv_stand and place items on the kitchen_counter
(64) check if there are bananas in the fridge; if not, get one from the
kitchen and put it in the fridge
(65) fetch a yellow box from the refrigerator and place it on the table, if
there isn't one, get it from the kitchen
(66) clean the sofa and put all items on the table in front of it
(67) find an apple and place it in the drawer
(68) Put the red bowl on the blue table in the fridge.
(69) go to the second floor
(70) keep the number of red_boxes on the yellow_table to 3 and put extra
red_boxes to the kitchen_counter
(71) clean up the beige table and put all items to kitchen
(72) put all fruit in the living_room to the fridge
(73) find an apple and place it in the fridge
(74) find a bowl and a mug then put them into the kitchen sink
(75) replenish the number of pears in the fridge to 3
(76) find an apple and put it on the brown table
(77) put all lemons and apples on the sofa to the tvstand
(78) put all bowls in the open drawer onto the kitchen_counter
(79) clean up the sofa and put all items into the drawer
(80) clean up the sofa and place all items on the nearby chair
(81) freeze the meat can on the blue desk
(82) check and close all appliances in the room
(83) get a cold apple and put it on the wood table
(84) check if there are anything in the kitchen drawer, if it's empty put a
short can into it
(85) turn off all appliances in the room then go the door and wait
(86) prepare some food and put it on the brown table
(87) check items in the fridge then increase the number of blue cans to 2
(88) find a box and put it on the tvstand
(89) clean the table in front of you and put all items into the sink
(90) find two bananas on the tvstand and put them to the kitchen sink
(91) find the bowl in the drawer and put it to the kitchen sink
```

```
(92) get a cold fruit and prepare to wash it
(93) clean the sofa
(94) put all items on the sofa to the tvstand
(95) put all items on the blue sofa to the white desk
(96) find the sponge and put it into the drawer
(97) find two kinds of fruit and put them on the tvstand
(98) find a banana and place it in a bowl
(99) put the bowl on the brown table into the kitchen sink and put the
suger_box on the tvstand to the sofa
(100) put the green_can on the brown_table to the fridge
```

## C  Metric Calculation

### C.1  Task Progress

In the task demonstrated in Appendix B, it's easy to find that to complete the task, we have to open the drawer to see if there is anything, and then we have to finish a put operation (put short can in the drawer). In addition to these two, we can also add some operation like, go to the drawer, close the cook and other actions which do not influence the final success. So we get the keypath as shown below,

```
[
    [
        "[Open, kitchen_drawer]",
        "[Put, short_can, kitchen_drawer]",
        "[End]"
    ],
    [
        "[Open, kitchen_drawer]",
        "[Put, short_can, kitchen_drawer]",
        "[Close, drawer]",
        "[End]"
    ],
    [
        "[Go to, drawer]",
        "[Open, kitchen_drawer]",
        "[Put, short_can, kitchen_drawer]",
        "[End]"
    ],
    [
        "[Go to, drawer]",
        "[Open, kitchen_drawer]",
        "[Put, short_can, kitchen_drawer]",
        "[Close, drawer]",
        "[End]"
    ]
]
```

Here's an example to show how to calculate TP,

```
(1) [Go to, kitchen](success)
(2) [Open, drawer](success)
(3) [Put, short_can, drawer](fail)
(4) [Go to, kitchen_counter](success)
(5) [Put, short_can, kitchen_counter](fail)
(6) [Go to, drawer](success)
(7) [Put, short_can, drawer](fail)
(8) [Go to, kitchen_counter](success)
```

```
(9)  [Put, short_can, kitchen_counter](fail)
(10) [Go to, drawer](success)
(11) [Put, short_can, drawer](fail)
(12) [Go to, kitchen_counter](success)
(13) [Put, short_can, kitchen_counter](fail)
(14) [Go to, drawer](success)
(15) [Put, short_can, drawer](fail)
(16) [Go to, kitchen_counter](success)
(17) [Put, short_can, kitchen_counter](fail)
(18) [Go to, drawer](success)
(19) [Put, short_can, drawer](fail)
(20)  [Go to, kitchen_counter](success)
```

This is the result of one run, and we can see that the TP of this run is as calculated in Section 3.3, $max$ $(\frac{1}{3}, \frac{1}{4}, \frac{1}{2}, \frac{2}{5}) = 0.5$.

## C.2  Success End Rate

In the above result, the number of steps reach 20, and there is no *End* action to terminate the task. Here's a example to show the success end.

```
(1)  [Go to, kitchen_counter](success)
(2)  [Go to, yellow_box](success)
(3)  [Pick, yellow_box](success)
(4)  [Go to, fridge](success)
(5)  [Put, yellow_box, fridge](fail)
(6)  [Open, fridge](fail)
(7)  [Go to, kitchen_counter](success)
(8)  [Put, yellow_box, kitchen_counter](success)
(9)  [Go to, fridge](success)
(10) [Open, fridge](success)
(11) [Go to, kitchen_counter](success)
(12) [Pick, yellow_box](success)
(13) [Go to, fridge](success)
(14) [Put, yellow_box, fridge](success)
(15) [Close, fridge](success)
(16) [End]
```

This is the result of one run for the task *take a yellow box and put it into the fridge*, and we can judge by its keypath that it complete the task successfully. It has *End* action, so the *End* is a success end which can be treated as one of the numerators when calculating SER in Section 3.3. In fact, as said in Section 3.3, successful task trajectory must have one end, but there maybe other unsuccessful task trajectories have ends, that's why we calculating SER.

## C.3  Success Re-plan Rate

First of all, the next action our agent takes after the previous action failed is called replan. Use the above subsection result as an example, and it's a successful task trajectory. In the step 5, the agent try to put the yellow box in the fridge but failed, and then, it try to open the fridge which can be treated as a success replan even though it failed again. Since the action *open fridge* is a meaningful action which can lead to the final success. It's one of the numerators when calculating SRR in Section 3.3. Also, in the first subsection for TP, the example is an unsuccessful task trajectory, so actions like *put short can drawer* are not success replan.

# D High Level Planning

In this section, we will should how the high-level planner described in Section 4.2 works step by step. A running demonstration of our HomieBot is shown in Fig. D4. To provide more intuitive understanding, we extract core sections from the original code and adapt them into a more general and easy-to-understand format to illustrate the process flow, this processing method is also applied to all subsequent code demonstrations. First, we provide the system information used in HomieBot, and all subsequent references to system information are consistent with what is provided here.

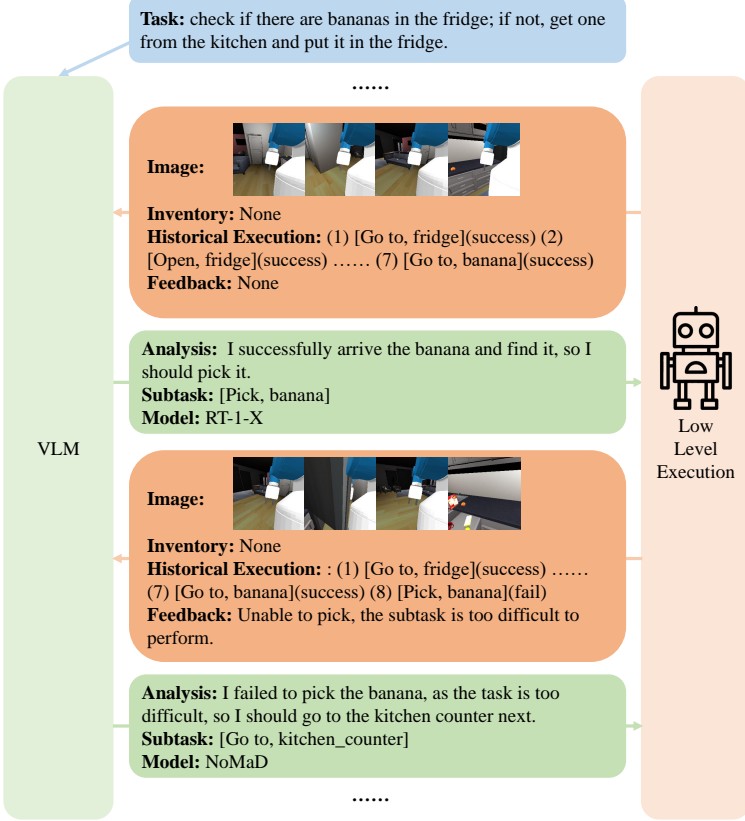

Figure D4: **An illustration of running pipeline of HomieBot.** After receiving images and feed- back, LMM generates analysis, specific subtask and model choice for low level executor to per- form.

```
You are a powerful housework assistant, I will give you following information
for you to make a decision toward the final task.
(1) Observation images: Four first-person perspective images of the current
environment, in the order of front, left, back, and right.
(2) Task: Your final goal.
(3) Inventory: Your current assets, remember that you are a one-hand agent,
which means you can't open or pick when your Inventory is not None, and you
can't put if your Inventory is None, this is very important.
(4) Historical Execution: Subtasks that were already fulfilled in the history,
 and the execution status of each subtask(success or fail). You need to make
decisions based on historical actions, current circumstances and your final
task.
(5) Feedback: Feedback will provide error information of the last execution,
it will be None if the last execution ends successfully.

You should output with following formats:
```

```
Analysis: Make a detailed summary of your current situation based on given
information, analyse and decide what to do next and output the reason of your
decision.
Subtask: [action, target], choose your action from the action list [Go to,
Pick, Put, Open, Close, End], and the target can be a place or a object from
your observation. If you choose Put as your action, output in format [Put,
object, place] which means put the object to the place. If the final task is
done and no more action is needed, just output [End].
Model: Choose one most suitable model in the model list [NoMaD, PixNav, octo,
RT-1-X]. NoMaD can go to a spot like living room, PixNav focuses on object
navigation and can go to a object, octo can handle with open and close, RT-1-X
 is good at picking and putting.

You need to focus on the consistency with previous subtasks. You should pay
attention to current Inventory and avoid conflicts.
Remember you can only go to the place and interact with the objects you
observe in your sight.
Remember the logic between outputs, it is recommended to open the receptacle
before you pick something because you can't open while holding, and it's
recommended to arrive the object place before you interact with it.
Remember you just need to output the next subtask to be fulfilled and don't
output a whole plan, this is very important.
Remember you should output strictly with the response template.
Now, I will send the message so that you can make planning accordingly.
```

Next, we define some classes to make the overall process more readable and smooth. Here we only list most relevant and important parts in the process.

```python
import os
import json
import re

class Conversations:
    def __init__(self, max_round=20):
        self.system = SYSTEM_INFO
        self.history = []
        self.round = 0
        self.window = 3
        self.max_round = max_round

    def get_history_prompt(self):
        history_prompt = ""
        if self.round < self.window:
            history_prompt = "".join(self.history)
        else:
            history_prompt = "".join(self.history[-3:])
        return history_prompt

    def reset(self):
        self.history = []
        self.round = 0

    def save(self, save_path):
        with open(os.path.join(save_path, "conversation.json"), "w") as file:
            json.dump(self.history, file, indent=4)

class HomieBot:
    def __init__(self):
        self.conv = Conversations()
```

```python
        self.inventory = []
        self.comm = Communicator()

    def get_inventory(self):
        if len(self.inventory) == 0:
            return "None"
        else:
            return " ".join(self.inventory)

    def generate_instruction(self, task, feedback, historical_execution):
        if historical_execution == "":
            instruction = f"Task: {task}\nInventory: {self.get_inventory()}\
            nHistorical Execution: None\nFeedback: None\nNow based on the
            instruction above, please output Analysis, Subtask and Model in
            mentioned format.\n"
        else:
            instruction = f"Task: {task}\nInventory: {self.get_inventory()}\
            nHistorical Execution: {historical_execution}\nFeedback: {feedback
            }\nNow based on the instruction above, please output Analysis,
            Subtask and Model in mentioned format.\n"
        return instruction

    def update_inventory(self, subtask, feedback):
        subtask = subtask.lower()
        if "None" in feedback:
            if "pick" in subtask:
                obj = subtask.split.split(',')[1].strip()
                self.inventory.append(obj)
            if "put" in subtask:
                self.inventory.pop()
        else:
            if "put" in subtask and "the object is missing" in feedback:
                self.inventory.pop()

    def end(self):
        self.comm.close_connection()
```

the most important function *generate_instruction* works as described in Section 4.2, which contains *task*, *inventory*, *history* and *feedback*.

Afterward, we provide the process for HomieBot to execute the task in a single trajectory.

```python
homie = HomieBot()
task = "input your task"
save_path = "save_path"
feedback = ""
historical_execution = ""

while homie.conv.round < homie.conv.max_round:
    homie.conv.round += 1
    instruction = homie.generate_instruction(task, feedback,
    historical_execution)
    images = homie.comm.receive_env_images()

    output = model_inference(instruction, images)
    homie.conv.history.append(f"USER:\n{instruction}ASSISTANT:\n{output}\n")

    pattern = r'.*Analysis: *(.+?) *Subtask: *\[(.*?)\].*Model: *(.*?)$'
    match = re.search(pattern, output, re.DOTALL)
```

```
    analysis = match.group(1).strip()
    subtask = match.group(2).strip()
    model_choice = match.group(3).strip()

    homie.comm.send_subtask(subtask, model_choice, homie.get_inventory())
    feedback, signal = homie.comm.receive_feedback()

    homie.update_inventory(subtask, feedback)
    historical_execution += f"({homie.conv.round}) {subtask}({signal}) "

    if "end" in subtask.lower():
        break

homie.conv.save(save_path)
homie.end()
```

the realization of function *model_inference* varies from different models, but it's quite easy to deploy different models into HomieBot as we can see in the code.

## E    Low Level Execution

### E.1    Pipeline

```
def error_detection(action, target, inventory, env):
    # Format Error Detection
    if action not in action_list:
        return 'fail', f'{action} is not in the action list! You should only
        choose actions in the list.'

    mapping_dict = load_name_mapping()
    if target in mapping_dict:
        target = mapping_dict[target]
    else:
        return 'fail', f'{target} does not exist! Please choose another object
        '

    # Logical Error Detection
    if inventory != 'None' and action in ['pick', 'open', 'close']:
        return 'fail', f'Unable to {action}, the hand is full'
    if inventory == 'None' and action == 'put':
        return 'fail', f'Unable to {action}, the hand is empty'

    if action == 'put' and "closed" in check_status(target):
        return 'fail', f'Unable to put, the {target} is closed, you should
        open it first'

    if action in ['open','close'] and "non-interactive" in check_status(target
    ):
        return 'fail', f'Can not {action} {target}! Please choose another
        object'

    # Distance Error Detection
    if action != "go to":
        distance = calculate_distance(env, target)
        if distance > 2:
            return 'fail', f'Unable to {action}, the target is far away'
        if distance < 0.1:
            return 'fail', f'Unable to {action}, the target is too close'
```

```python
    return 'success', 'None'

max_count = 20
comm = Communicator()
save_path = "save_path"
count_steps = 1
env = init_env()

while count_steps <= max_count:
    images = get_env_images(save_path, env, count_steps)
    comm.send_env_images(images)

    action, target, inventory = comm.receive_subtask()
    if "end" in action.lower():
        comm.send_feedback("None", "success")
        break

    # Error Detection Before Execution
    signal, feedback = error_detection(action, target, inventory, env)
    if signal == "fail":
        comm.send_feedback(feedback, signal)
        break

    for retry in range(3):
        reset_arm(env)
        # Error Detection During and After Execution
        signal, feedback, env = execution(action, target, inventory, env)
        if signal == 'success':
            break
        elif action == 'put' and env['grasped_obj'] is None:
            feedback = f'Unable to {action}, and the object is missing'
            break
        elif retry == 2:
            feedback = f'Unable to {action}, the subtask is too difficult to
            perform'
    if signal == 'success':
        feedback = "None"

    count_steps += 1
    comm.send_feedback(feedback, signal)
```

### E.2    Skills

The skill we choose and their functions are shown in Table E1.

### E.3    Models

**M3**   (Gu et al., 2022) can flexible interact with target objects from various locations based on the integration of manipulative skills and mobility, while navigational skills are designed to accommodate multiple endpoints, ultimately leading to successful operations. Specifically, M3 implements these concepts by emphasizing mobile manipulation skills over fixed skills and training navigational skills using area targets rather than point targets.

**RT-1-X**   ( (Padalkar et al., 2023)) architecture utilizes image and text instructions as inputs, and generates discrete end-effector actions as outputs. Specifically, RT-1-X is a transformer-based model that guides robotic

Table E1: The list of skills we used with descriptions and examples

| SKILL | DESCRIPTION | EXAMPLE |
|---|---|---|
| PICK OBJECT | PICK AN OBJECT UP | PICK SUGAR BOX |
| PUT OBJECT TO PLACE | PUT AN OBJECT INTO A PLACE | PUT LEMON ON BROWN TABLE |
| OPEN CONTAINER | OPEN THE CONTAINER | OPEN THE FRIDGE |
| CLOSE CONTAINER | CLOSE THE CONTAINER | CLOSE THE KITCHEN DRAWER |
| GO TO PLACE | NAVIGATE TO A PLACE | NAVIGATE TV STAND |
| GO TO OBJECT | NAVIGATE TO WHERE AN OBJECT IS | NAVIGATE BOWL |
| END | END THE EXECUTION | END |

Table E2: Descriptions of Low Level Models used in HOMIEBOT.

| MODEL | INPUT | CAPABILITY | TASK |
|---|---|---|---|
| RT-1-X (BROHAN ET AL., 2022) | RGB & INSTRUCTIONS | MANIPULATION | PICKING & PLACING |
| OCTO (TEAM ET AL., 2024B) | RGB & INSTRUCTIONS | MANIPULATION | OPENING & CLOSING |
| NOMAD (SRIDHAR ET AL., 2024) | RGB & GOAL-IMAGE | IMAGE-NAVIGATION | NAVIGATE TO SPOT & LARGE OBJECT |
| PIXNAV (CAI ET AL., 2024) | RGB & GOAL-NAME | PIXEL-NAVIGATION | NAVIGATE TO OBJECT |

arms to complete various manipulation tasks. RT-1-X is an extension of the RT-1 ( (Brohan et al., 2022))
model, which is designed for robot control and trained on a large-scale robot dataset.

**Octo** ( (Team et al., 2024b)) is an open-source, general-purpose policy for robotic manipulation based on
transformers. It supports flexible task and observation definition and can be quickly integrated into new
observation and action spaces.

**NoMaD** ( (Sridhar et al., 2024)) trains a single diffusion strategy for goal-oriented navigation and goal-
independent exploration, the first one is to reach user-specified goals after localization and the second one is
to search new environments. The method is instantiated using a transformer-based large-scale policy trained
on data from various ground robots.

**PixNav** ( (Cai et al., 2024)) is a pixel-guided navigational skill. It designs an LLM-based planner that
utilizes common sense between objects and rooms to select the optimal waypoints, which are then executed
by a pixel navigation strategy to achieve long-line-of-sight navigation. In this pipeline, we use its ability of
finding the optimal waypoint and pixel navigation to navigate to some specific small object such as lemon
and sugar box.

### E.4  Error Classification

**Logical error**   If the hand already has an object (inventory is not empty) but still attempts to perform a
pick/open/close operation, the execution will fail, and the message *the hand is full* will be returned; if the
hand has no object (inventory is empty) but still attempts to perform a place operation, the execution will
fail, and the message *the hand is empty* will be returned; if the item is not a container but still attempts to
perform a open/close operation, the execution will fail, and the message *please choose another object* will
be returned. In the execution with environment state information, if the container is closed and a place
operation is still attempted, the execution will fail, and the message *the container is closed, you should open
it first* will be returned.

**Distance error**   In the execution with environment state information, if the agent is too close to the target,
causing the arm to be unable to extend properly but still attempts to perform a pick/place/open/close

operation, the execution will fail, and the message *the target is too close* will be returned; if the agent is too far from the target, causing it to be unable to reach the target object but still attempts to perform a pick/place/open/close operation, the execution will fail, and the message *the target is far away* will be returned.

**Format Error**   For high level planning, it may output an object which is not in the scene, that is, in low level execution, we can't find an object with a name matching the input in the scene, the message *please choose another object* will be returned; also, high level planning may output in a wrong operation which can not be performed, the message *You should only choose actions in the list* will be returned.

**Execution Error**   Due to the limited capabilities of low-level models, sometimes the failure is not caused by HLP. Therefore, each action can be executed up to three times. If it fails after three times, it will return a message *the subtask is too difficult to perform*; also, when performing a put operation, if the agent put the wrong place, it will return a message *the object is missing* to remind the agent to re-plan and re-pick.

# F   Data Augmentation

## F.1   SFT Augmentation

To expand the original dataset size, we first use GPT-4o (Achiam et al., 2023) to regenerate text descriptions. Here is the regeneration code clip, we just show how to regenerate task descriptions, but the regeneration of subtask analysis uses the same template.

```python
client = OpenAI(api_key='')
completion = client.chat.completions.create(
    model="gpt-4o",
    messages=[
        {"role": "system", "content": "Rewrite the following text with the
        same meaning but in a different description while do not change object
        's name: "},
        {"role": "user", "content": task}
    ]
)
```

Next we show how to convert a single EMMOE data into fix-format conversation data. After processing, each individual subtask will be combined with all previously subtasks to form a SFT data.

```python
import os
with open(task_path) as file:
    content = file.read()

content = content.split("\n\n")
task = content[0]
historical = ""
sft_data = []

for i, subtask_info in enumerate(content[1:]):
    subtask_data = {}
    subtask_info = subtask_info.strip().split("\n")
    if subtask_info[0] == '':
        continue
    subtask_id, decision = subtask_info[0].split(': ')
    subtask_id = subtask_id.lower()
    analysis = subtask_info[1]

    if "End" not in decision:
        action, model_choice = decision.strip(')').split(' (')
```

```python
    else:
        action = "[End]"
        model_choice = "None"

    image_paths = [
        os.path.join(save_dir, f"{subtask_id}_front.png"),
        os.path.join(save_dir, f"{subtask_id}_left.png"),
        os.path.join(save_dir, f"{subtask_id}_back.png"),
        os.path.join(save_dir, f"{subtask_id}_right.png")
    ]
    for path in image_paths:
        if not os.path.exists(path):
            raise FileNotFoundError(f"File does NOT exist: {path}")
    if i == 0:
        instruction = f"{task}\nInventory: None\nHistorical Execution: None\
        nFeedback: None\nNow, please output Analysis, Subtask and Model,\
        according to the instruction above."
    else:
        instruction = f"{task}\n{inventory}\nHistorical Execution:{historical\
        }\n{feedback}\nNow, please output Analysis, Subtask and Model,\
        according to the instruction above."
    answer = f"{analysis}\nSubtask: {action}\nModel: {model_choice}"

    feedback = subtask_info[2]
    inventory = subtask_info[3]
    if "None" in feedback:
        historical += f"({i+1}){decision} (success)\n"
    else:
        historical += f"({i+1}){decision} (fail)\n"

    conv = []
    conv.append({"from": "human", "value": instruction})
    conv.append({"from": "gpt", "value": answer})

    task_id = '_'.join(task_path.split('.')[0].split('/')[-2:])

    subtask_data["id"] = task_id + '_' + subtask_id
    subtask_data["image"] = image_paths
    subtask_data["conversations"] = conv

    sft_data.append(subtask_data)
```

We also provide some data samples for more intuitive understanding.

```json
[
    {
        "id": "21_info_subtask13",
        "image": [
            "./HomieBot/data/train/21/subtask13_front.png",
            "./HomieBot/data/train/21/subtask13_left.png",
            "./HomieBot/data/train/21/subtask13_back.png",
            "./HomieBot/data/train/21/subtask13_right.png"
        ],
        "conversations": [
            {
                "from": "human",
                "value": "Task: clean up the blue sofa\nInventory: None\
                nHistorical Execution:(1)[Go to, blue_sofa] (PixNav) (success)
                \n(2)[Pick, cracker_box] (RT-1-X) (success)\n(3)[Go to,
```

```
                kitchen_counter] (NoMaD) (success)\n(4)[Put, cracker_box,
                kitchen_counter] (RT-1-X) (success)\n(5)[Go to, blue_sofa] (
                PixNav) (success)\n(6)[Pick, sugar_box] (RT-1-X) (success)\n
                (7)[Go to, kitchen_counter] (NoMaD) (success)\n(8)[Put,
                sugar_box, kitchen_counter] (RT-1-X) (success)\n(9)[Go to,
                blue_sofa] (PixNav) (success)\n(10)[Pick, soup_can] (RT-1-X) (
                success)\n(11)[Go to, kitchen_counter] (NoMaD) (success)\n(12)
                [Put, soup_can, kitchen_counter] (RT-1-X) (success)\n\
                nFeedback: None\nNow, please output Analysis, Subtask and
                Model, according to the instruction above."
        },
        {
                "from": "gpt",
                "value": "Analysis: I successfully put the soup_can on the
                kitchen_counter, according to my task, I should go back to the
                 sofa to check whether it's clean or not\nSubtask: [Go to,
                blue_sofa]\nModel: PixNav"
        }
    ]
},
{
    "id": "21_info_subtask14",
    "image": [
        "./HomieBot/data/train/21/subtask14_front.png",
        "./HomieBot/data/train/21/subtask14_left.png",
        "./HomieBot/data/train/21/subtask14_back.png",
        "./HomieBot/data/train/21/subtask14_right.png"
    ],
    "conversations": [
        {
                "from": "human",
                "value": "Task: clean up the blue sofa\nInventory: None\
                nHistorical Execution:(1)[Go to, blue_sofa] (PixNav) (success)
                \n(2)[Pick, cracker_box] (RT-1-X) (success)\n(3)[Go to,
                kitchen_counter] (NoMaD) (success)\n(4)[Put, cracker_box,
                kitchen_counter] (RT-1-X) (success)\n(5)[Go to, blue_sofa] (
                PixNav) (success)\n(6)[Pick, sugar_box] (RT-1-X) (success)\n
                (7)[Go to, kitchen_counter] (NoMaD) (success)\n(8)[Put,
                sugar_box, kitchen_counter] (RT-1-X) (success)\n(9)[Go to,
                blue_sofa] (PixNav) (success)\n(10)[Pick, soup_can] (RT-1-X) (
                success)\n(11)[Go to, kitchen_counter] (NoMaD) (success)\n(12)
                [Put, soup_can, kitchen_counter] (RT-1-X) (success)\n(13)[Go
                to, blue_sofa] (PixNav) (success)\n\nFeedback: None\nNow,
                please output Analysis, Subtask and Model, according to the
                instruction above."
        },
        {
                "from": "gpt",
                "value": "Analysis: According to the historical execution and
                final task, I have finally finished the task and it's time to
                end\nSubtask: [End]\nModel: None"
        }
    ]
}
]
```

### F.2 DPO Augmentation

The last step for SFT Augmentation and the first step of DPO Augmentation occur simultaneously, failed subtasks are considered junk data for SFT but can be used to construct DPO data, so we run *data_sift* function first. After getting initial DPO data, we will use SFT data, *model_change*, *order_change* and *action_change* to construct final DPO dataset.

```python
def data_sift(subtask_list):
    sft_data = []
    dpo_data = []
    flag = 1
    for i in range(1, len(subtask_list)):
        if "Feedback: None" in subtask_list[i]["conversations"][0]["value"]:
            sft_data.append(subtask_list[i-1])
            if flag == 0:
                dpo_data.append({
                    "prompt": subtask_list[i-2]["conversations"][0]["value"],
                    "chosen": '\n'.join(subtask_list[i-1]["conversations"][1][
                    "value"].split('\n')[1:]),
                    "rejected": '\n'.join(subtask_list[i-2]["conversations"
                    ][1]["value"].split('\n')[1:])
                })
                flag = 1
        else:
            flag = 0
    sft_data.append(subtask_list[-1])

    return sft_data, dpo_data

def dpo_augment(sft_data, dpo_data):
    for i in range(len(sft_data)):
        prompt = sft_data[i]["conversations"][0]["value"]
        chosen = '\n'.join(sft_data[i]["conversations"][1]["value"].split('\n'
        )[1:])
        if "End" in sft_data[i]["conversations"][1]["value"]:
            continue

        def model_change(chosen):
            if "NoMaD" in chosen:
                return chosen.replace("NoMaD", "PixNav")
            elif "PixNav" in chosen:
                return chosen.replace("PixNav", "NoMaD")
            elif "octo" in chosen:
                return chosen.replace("octo", "RT-1-X")
            else:
                return chosen.replace("RT-1-X", "octo")

        def order_change(i, sft_data):
            return '\n'.join(sft_data[i+1]["conversations"][1]["value"].split(
            '\n')[1:])

        def action_change(chosen):
            if "Pick" in chosen:
                return chosen.replace("Pick", "Fetch")
            elif "Put" in chosen:
                return chosen.replace("Put", "Place")
            elif "Go to" in chosen:
                return chosen.replace("Go to", "Move")
            elif "Open" in chosen:
```

```
                return chosen.replace("Open", "Pull")
            elif "Close" in chosen:
                return chosen.replace("Close", "Push")

        reject1 = model_change(chosen)
        reject2 = order_change(i, sft_data)
        reject3 = action_change(chosen)
        dpo_data.append({"prompt": prompt, "chosen": chosen, "rejected":
        reject1})
        dpo_data.append({"prompt": prompt, "chosen": chosen, "rejected":
        reject2})
        dpo_data.append({"prompt": prompt, "chosen": chosen, "rejected":
        reject3})

    return dpo_data
```

Notably, action $End$ is special among all available actions and it will only appear as $rejected$ in DPO data. In the first augmentation stage and $order\_change$, since the relationship between $chosen$ and $rejected$ is $O_i$ and $O_{i+1}$ (see definitions in Section 5.1) and there are no other subtasks after $End$, which means other actions might appear in either $chosen$ or $rejected$ while $End$ can only be the $rejected$. But this effect of suppressing the $End$ output is exactly what we want. Even executing a few extra steps after completing the task is better than terminating early without finishing the task. That is to say, We hope the model could consider more and do not output $End$ so easily. Experimental results in Table 2 and Table 3 confirm the effectiveness of this method as we can see an improvement in $SER$ metric, another positive phenomenon in results is that the length of the successful paths hasn't increased significantly as we observe in $PLWSR$ and $TP$.

Finally, we provide some DPO data examples.

```
[
    {
        "prompt": "Task: Clear everything off the table in front of you and
        place all the items in the sink.\nInventory: None\nHistorical
        Execution:(1)[Pick, yellow_box] (RT-1-X) (success)\n(2)[Put,
        yellow_box, sink] (RT-1-X) (success)\n\nFeedback: None\nNow, please
        output Analysis, Subtask and Model, according to the instruction above
        .",
        "chosen": "Subtask: [Go to, red_can]\nModel: PixNav",
        "rejected": "Subtask: [Pick, red_can]\nModel: RT-1-X"
    },
    {
        "prompt": "Task: Collect all the fruit located on the brown table and
        place them on the sofa.\nInventory: None\nHistorical Execution:(1)[Go
        to, brown_table] (NoMaD) (success)\n(2)[Pick, orange] (RT-1-X) (
        success)\n(3)[Go to, sofa] (PixNav) (success)\n(4)[Put, orange, sofa]
        (RT-1-X) (success)\n(5)[Go to, brown_table] (NoMaD) (success)\n\
        nFeedback: None\nNow, please output Analysis, Subtask and Model,
        according to the instruction above.",
        "chosen": "Subtask: [Pick, pear]\nModel: RT-1-X",
        "rejected": "Subtask: [Fetch, pear]\nModel: RT-1-X"
    },
    {
        "prompt": "Task: find a blue can for me\nInventory: None\nHistorical
        Execution: None\nFeedback: None\nNow, please output Analysis, Subtask
        and Model, according to the instruction above.",
        "chosen": "Subtask: [Go to, fridge]\nModel: PixNav",
        "rejected": "Subtask: [Go to, fridge]\nModel: NoMaD"
    }
]
```

## G  Training Details

### G.1  Training Parameters

We use Video-LLaVA-7B ([Zhang et al., 2023](#)) as our base model, we also use the training scripts they provide and partial parameters for $sft$ are as follows.

```
--lora_enable True
--lora_r 128
--lora_alpha 256
--mm_projector_lr 2e-5
--bits 4
--mm_projector_type mlp2x_gelu
--mm_vision_select_layer -2
--mm_use_im_start_end False
--mm_use_im_patch_token False
--image_aspect_ratio pad
--group_by_modality_length True
--bf16 True
--num_train_epochs 1
--per_device_train_batch_size 16
--per_device_eval_batch_size 4
--gradient_accumulation_steps 1
--evaluation_strategy "no"
--save_strategy "steps"
--save_steps 50000
--save_total_limit 1
--learning_rate 5e-4
--weight_decay 0.
--warmup_ratio 0.03
--lr_scheduler_type "cosine"
--logging_steps 1
--tf32 True
--model_max_length 2048
--tokenizer_model_max_length 3072
--gradient_checkpointing True
--dataloader_num_workers 4
--lazy_preprocess True
--report_to tensorboard
```

We use finetuned model as our base and reference model, and use open-source $trl$ package and parameters for $dpo$ are as follows.

```
bnb_config = BitsAndBytesConfig(
    load_in_4bit=True,
    bnb_4bit_compute_dtype=torch.float16,
    bnb_4bit_use_double_quant=True,
    bnb_4bit_quant_type='nf4'
)
training_args = DPOConfig(
    per_device_train_batch_size=16,
    per_device_eval_batch_size=4,
    gradient_accumulation_steps=1,
    gradient_checkpointing=True,
    max_grad_norm=0.3,
    num_train_epochs=1,
```

```
    save_steps=1000,
    learning_rate=5e-6,
    bf16=True,
    save_total_limit=1,
    logging_steps=10,
    output_dir=output_dir,
    optim="paged_adamw_32bit",
    lr_scheduler_type="cosine",
    warmup_ratio=0.03,
    remove_unused_columns=False
)
peft_config = LoraConfig(
    r=8,
    lora_alpha=8,
    target_modules=find_all_linear_names(model),
    lora_dropout=0.05,
    bias="none",
    task_type="CAUSAL_LM",
)
dpo_trainer = DPOTrainer(
    model,
    model_ref,
    args=training_args,
    beta=0.1,
    train_dataset=train_dataset,
    eval_dataset=eval_dataset,
    tokenizer=tokenizer,
    max_prompt_length=2048,
    max_length=2048,
)
```

# H  Experimental Details

## H.1  Baseline Setup

To make it more convenient for different models to deploy into our system without training, we slightly lower output format requirements, here shows the adapatations.

```
import re

pattern = r'.*Analysis: *(.+?) *Subtask: *\[(.*?)\].*Model: *(.*?)$'
match = re.search(pattern, output, re.DOTALL)
if match == None:
    pattern = r'.*Analysis: *(.+?) *Subtask: *(.*?) *Model: *(.*?)$'
    match = re.search(pattern, output, re.DOTALL)
```

Despite lowering the output format standards, the output from 7B-sized models still fails to meet our least requirements. They either do not output single-step subtasks or the subtask format is far from requirements. This issue is difficult to resolve by merely adjusting prompts. Therefore, we leverage the in-context learning abilities of these models by providing an output template example before each inference. Here, we provide the inference template for Qwen2-VL (Wang et al., 2024b) MiniCPM-V 2.6 (Yao et al., 2024) respectively.

Qwen2VL

```
messages = [
        {"role": "system", "content": homie.conv.system},
        {"role": "user",
```

```
            "content": "here is an example output, please strictly follow its
            format and system reminders in your output:\nAnalysis: According to
            my final task, I need to fetch apples first, but it's a better choice
             to go the fridge and open it first, which will avoid potential
            conflicts, so I should go to the fridge next\nSubtask: [Go to, fridge
            ]\nModel: NoMaD\n",
        },
        {"role": "assistant",
         "content": "I will surely follow the given format, now you can send
         prompt to me."
        },
        {"role": "user",
         "content": [
                {"type": "image", "image": images[0]},
                {"type": "image", "image": images[1]},
                {"type": "image", "image": images[2]},
                {"type": "image", "image": images[3]},
                {"type": "text", "text": instruction}]
        }
]
prompt = processor.apply_chat_template(
        messages, tokenize=False, add_generation_prompt=True
)
image_inputs, video_inputs = process_vision_info(messages)
inputs = processor(
        text=[prompt],
        images=image_inputs,
        videos=video_inputs,
        padding=True,
        return_tensors="pt"
).to("cuda")
generated_ids = model.generate(**inputs, max_new_tokens=512)
enerated_ids_trimmed = [
        out_ids[len(in_ids) :] for in_ids, out_ids in zip(inputs.input_ids,
        generated_ids)
]
outputs = processor.batch_decode(
        generated_ids_trimmed, skip_special_tokens=True,
        clean_up_tokenization_spaces=False
)
```

MiniCPM-V 2.6

```
image_loads = [Image.open(image).convert('RGB') for image in images]
messages = [
        {"role": "user",
         "content": "here is an example output, please strictly follow its
         format and system reminders in your output:\nAnalysis: According to
         my final task, I need to fetch apples first, but it's a better choice
          to go the fridge and open it first, which will avoid potential
         conflicts, so I should go to the fridge next\nSubtask: [Go to, fridge
         ]\nModel: NoMaD\n",
        },
        {"role": "assistant",
         "content": "I will surely follow the given format, now you can send
         prompt to me.",
        },
        {"role": "user",
```

```
            "content": [image_loads[0], image_loads[1], image_loads[2],
            image_loads[3], instruction]
        }
]

output = model.chat(
        image=None,
        system_prompt=homie.conv.system,
        tokenizer=tokenizer
)
```

## H.2 Results

Here we provide more detailed results of experiments in Section 5.5. Table H3 and Table H4 show the statistics results in percentages while Table H5 and Table H6 show original counts. Table H7 show the original counts and success rate range of each action.

Table H3: **Successful Trajectories Error Statistics** All definitions are same as in Section 5.5. Additionally, we add statistics of four primary types.

| Models | L1 | L2 | L3 | L4 | L | D1 | D2 | D | F1 | F2 | F | E1 | E2 | E | All |
|---|---|---|---|---|---|---|---|---|---|---|---|---|---|---|---|
| GPT-4o (ACHIAM ET AL., 2023) | 3.97 | 0.79 | 0.79 | 0 | 5.56 | 44.44 | 0 | 44.44 | 1.59 | 17.46 | 19.05 | 15.87 | 15.08 | 30.95 | 30.29 |
| GEMINI-1.5-PRO (TEAM ET AL., 2024A) | 3.85 | 3.85 | 0 | 7.69 | 15.38 | 48.08 | 0 | 48.08 | 0 | 17.31 | 17.31 | 15.38 | 3.85 | 19.23 | 21.80 |
| QWEN2-VL-7B (WANG ET AL., 2024B) | 0 | 0 | 0 | 0 | 0 | 100 | 0 | 100 | 0 | 0 | 0 | 0 | 0 | 0 | 20 |
| MINICPM-V 2.6 (YAO ET AL., 2024) | 0 | 0 | 0 | 0 | 0 | 100 | 0 | 100 | 0 | 0 | 0 | 0 | 0 | 0 | 6.67 |
| o1 (JAECH ET AL., 2024) | 0 | 0 | 1.18 | 10.06 | 11.24 | 21.89 | 0 | 21.89 | 0 | 48.52 | 48.52 | 18.34 | 0 | 18.34 | 20.56 |
| HOMIEBOT-7B (SFT) | 10.53 | 9.77 | 12.78 | 1.50 | 34.59 | 36.09 | 0 | 36.09 | 0 | 3.01 | 3.00 | 24.06 | 2.26 | 26.32 | 14.41 |
| HOMIEBOT-7B (SFT+DPO) | 10.17 | 15.25 | 9.32 | 3.39 | 38.14 | 33.05 | 0 | 33.05 | 0 | 3.39 | 3.39 | 25.42 | 0 | 25.42 | 12.87 |

Table H4: **Failed Trajectories Error Statistics**

| Models | L1 | L2 | L3 | L4 | L | D1 | D2 | D | F1 | F2 | F | E1 | E2 | E | All |
|---|---|---|---|---|---|---|---|---|---|---|---|---|---|---|---|
| GPT-4o (ACHIAM ET AL., 2023) | 6.87 | 0.12 | 0.69 | 3.65 | 11.34 | 8.41 | 0.06 | 8.47 | 0.57 | 64.88 | 65.45 | 13.99 | 0.75 | 14.74 | 73.61 |
| GEMINI-1.5-PRO (TEAM ET AL., 2024A) | 7.48 | 1.52 | 2.41 | 6.45 | 17.86 | 9.41 | 0 | 9.41 | 0 | 47.86 | 47.86 | 22.76 | 2.10 | 24.86 | 68.38 |
| QWEN2-VL-7B (WANG ET AL., 2024B) | 2.17 | 9.49 | 0.99 | 3.56 | 16.21 | 7.71 | 0 | 7.71 | 4.74 | 54.35 | 59.09 | 16.40 | 0.59 | 17.00 | 27.74 |
| MINICPM-V 2.6 (YAO ET AL., 2024) | 8.58 | 0.80 | 0.92 | 1.72 | 12.01 | 7.78 | 0 | 7.78 | 3.49 | 65.39 | 68.88 | 10.87 | 0.46 | 11.33 | 31.08 |
| o1 (JAECH ET AL., 2024) | 1.16 | 0.07 | 1.96 | 4.86 | 8.05 | 8.64 | 0 | 8.64 | 0 | 56.75 | 56.75 | 26.27 | 0.29 | 26.59 | 51.77 |
| HOMIEBOT-7B (SFT) | 11.31 | 23.85 | 9.86 | 4.20 | 49.24 | 11.77 | 0 | 11.77 | 0.61 | 11.47 | 12.08 | 24.54 | 2.37 | 26.91 | 35.70 |
| HOMIEBOT-7B (SFT+DPO) | 11.46 | 23.90 | 11.13 | 2.62 | 49.10 | 9.25 | 0 | 9.25 | 0.25 | 17.27 | 17.51 | 22.67 | 1.47 | 24.14 | 35.88 |

Table H5: **Original Successful Trajectories Statistics** All data are integers.

| Models | L1 | L2 | L3 | L4 | L | D1 | D2 | D | F1 | F2 | F | E1 | E2 | E | All |
|---|---|---|---|---|---|---|---|---|---|---|---|---|---|---|---|
| GPT-4o (ACHIAM ET AL., 2023) | 5 | 1 | 1 | 0 | 7/126 | 56 | 0 | 56/126 | 2 | 22 | 24/126 | 20 | 19 | 39/126 | 126/416 |
| GEMINI-1.5-PRO (TEAM ET AL., 2024A) | 4 | 4 | 0 | 8 | 16/104 | 50 | 0 | 50/104 | 0 | 18 | 18/104 | 16 | 4 | 20/104 | 104/477 |
| QWEN2-VL-7B (WANG ET AL., 2024B) | 0 | 0 | 0 | 0 | 0/9 | 9 | 0 | 9/9 | 0 | 0 | 0/9 | 0 | 0 | 0/9 | 9/45 |
| MINICPM-V 2.6 (YAO ET AL., 2024) | 0 | 0 | 0 | 0 | 0/1 | 1 | 0 | 0/1 | 0 | 0 | 0/1 | 0 | 0 | 0/1 | 1/15 |
| o1 (JAECH ET AL., 2024) | 0 | 0 | 2 | 17 | 19/169 | 37 | 0 | 37/169 | 0 | 82 | 82/169 | 31 | 0 | 31/169 | 169/822 |
| HOMIEBOT-7B (SFT) | 14 | 13 | 17 | 2 | 46/133 | 48 | 0 | 48/133 | 0 | 4 | 4/133 | 32 | 3 | 35/133 | 133/923 |
| HOMIEBOT-7B (SFT+DPO) | 12 | 18 | 11 | 4 | 45/118 | 39 | 0 | 39/118 | 0 | 4 | 4/118 | 30 | 0 | 30/118 | 118/917 |

# I Case Study

We show case studies of the inference from HomieBot in various situations as follows.

**Case 1: Successful trajectory**

Table H6: **Original Failed Trajectories Statistics**

| Models | L1 | L2 | L3 | L4 | L | D1 | D2 | D | F1 | F2 | F | E1 | E2 | E | All |
|---|---|---|---|---|---|---|---|---|---|---|---|---|---|---|---|
| GPT-4o (ACHIAM ET AL., 2023) | 228 | 4 | 23 | 121 | 376/3317 | 279 | 2 | 281/3317 | 19 | 2152 | 2171/3317 | 464 | 25 | 489/3317 | 3317/4506 |
| GEMINI-1.5-PRO (TEAM ET AL., 2024A) | 217 | 44 | 70 | 187 | 518/2900 | 273 | 0 | 273/2900 | 0 | 1388 | 1388/2900 | 660 | 61 | 721/2900 | 2900/4241 |
| QWEN2-VL-7B (WANG ET AL., 2024B) | 33 | 144 | 15 | 54 | 246/1518 | 117 | 0 | 117/1518 | 72 | 825 | 897/1518 | 249 | 9 | 258/1518 | 1518/5472 |
| MINICPM-V 2.6 (YAO ET AL., 2024) | 150 | 14 | 16 | 30 | 210/1748 | 136 | 0 | 136/1748 | 61 | 1143 | 1204/1748 | 190 | 8 | 198/1748 | 1748/5624 |
| HOMIEBOT-7B (SFT) | 148 | 312 | 129 | 55 | 644/1308 | 154 | 0 | 154/1308 | 8 | 150 | 158/1308 | 321 | 31 | 352/1308 | 1308/3664 |
| HOMIEBOT-7B (SFT+DPO) | 140 | 292 | 136 | 32 | 600/1222 | 113 | 0 | 113/1222 | 3 | 211 | 214/1222 | 277 | 18 | 295/1222 | 1222/3406 |

Table H7: *Count* represents the number of each action error with a total count behind. $SR_{range}$ is the percentage to indicate the range of success rates of each action, with the average value shown in parentheses.

| METRICS | GO TO | PICK | PLACE | OPEN | CLOSE |
|---|---|---|---|---|---|
| COUNT | 938/2437 | 1213/2437 | 178/2437 | 81/2437 | 27/2437 |
| $SR_{range}$ | $(45.32)31.19\sim82.89$ | $(22.45)18.34\sim28.95$ | $(40.97)29.06\sim69.40$ | $(43.13)30.13\sim75.84$ | $(36.45)26.72\sim57.37$ |

Here we show a successful trajectory of our HomieBot (DPO version). To facilitate understanding, we convert the dialogue data into the original EMMOE data format. As shown, even if errors occur, timely adjustments can be made through feedback, ensuring the correctness of the execution process.

```
Task: put the green_can on the brown_table to the fridge

#Round1
Analysis: According to my final task, I need to go to the green_can first.
Subtask: [Go to, green_can]
Inventory: None
Feedback: None

#Round2
Analysis: I successfully arrived the green_can and I should pick it next
Subtask: [Pick, green_can]
Inventory: green_can
Feedback: None

#Round3
Analysis: I successfully picked the green_can, so according to my final task,
I should go to the fridge to put it down.
Subtask: [Go to, fridge]
Inventory: green_can
Feedback: None

#Round4
Analysis: I successfully arrived the fridge and I should put the green_can
into it next
Subtask: [Put, green_can, fridge]
Inventory: green_can
Feedback: Unable to put, the fridge is closed, you should open it first

#Round5
Analysis: I failed to put the green_can into the fridge, since the fridge is
closed, so I should open the fridge first
Subtask: [Open, fridge]
Inventory: green_can
Feedback: Unable to open, the hand is full

#Round6
```

```
Analysis: I failed to open the fridge because my hand is full, so I should put
 the green_can on the brown_table first
Subtask: [Go to, brown_table]
Inventory: green_can
Feedback: None

#Round7
Analysis: I successfully arrive the brown_table and I should put the green_can
 on it
Subtask: [Put, green_can, brown_table]
Inventory: None
Feedback: None

#Round8
Analysis: I successfully put the green_can on the brown_table, so I should go
to the fridge next
Subtask: [Go to, fridge]
Inventory: None
Feedback: None

#Round9
Analysis: I successfully arrived the fridge and I should open it next
Subtask: [Open, fridge]
Inventory: None
Feedback: None

#Round10
Analysis: I successfully open the fridge and find the green_can, so I should
go to the green_can.
Subtask: [Go to, green_can]
Inventory: None
Feedback: None

#Round11
Analysis: I successfully arrive the green_can and find it, so I should pick it
.
Subtask: [Pick, green_can]
Inventory: green_can
Feedback: None

#Round12
Analysis: I successfully picked the green_can, so according to my final task,
I should go to the fridge to put it down.
Subtask: [Go to, fridge]
Inventory: green_can
Feedback: None

#Round13
Analysis: I successfully arrived the fridge and I should put the green_can
into it next
Subtask: [Put, green_can, fridge]
Inventory: None
Feedback: None

#Round14
Analysis: I successfully put the green_can into the fridge, so I should close
the fridge next
Subtask: [Close, fridge]
Inventory: None
```

```
Feedback: None\n

#Round15
Analysis: According to the historical execution and final task, I have finally
 finished the task and it's time to end
Subtask: [End]
Inventory: None
Feedback: None
```

**Case 2: Terrible Grounding**

Here we show a terrible grounding problem issues during the inference of GPT4-o (Achiam et al., 2023). Due to the lengthy path, we only highlight the most critical subtask outputs and their execution results to emphasize errors (following cases are handled similarly). We can see that the ungrounded output directly prevents the process to be continued. Even after informing the model that the object doesn't exist, the issue remains unresolved. The model continues to output incorrect objects or makes mistakes again after a few steps.

```
Task: clean up the beige table and put all items to kitchen

(1) Go to, beige table(success)
(2) Pick, item(fail)
(3) Pick, item(fail)
(4) Pick, item(fail)
(5) Pick, item(fail)
(6) Pick, item(fail)
(7) Pick, item(fail)
(8) Pick, item on the table(fail)
(9) Pick, item on the table(fail)
(10) Pick, item on the table(fail)
(11) Pick, item on the table(fail)
(12) Pick, item(fail)
(13) Pick, item on the table(fail)
(14) Pick, item on the beige table(fail)
(15) Pick, stack of papers(fail)
(16) Pick, box(fail)
(17) Go to, closer to the table(fail)
(18) Pick, item on the table(fail)
(19) Pick, visible item on the beige table(fail)
```

**Case 3: Limited LLE**

Here we show a problem issues from the limited ability of low-level models. As we can see in step7 and step8, though high-level planner makes correct plans, execution still fails due to the limited ability of low-level models, this problem occurs repeatedly during whole process and finally leads to the failue the trajectory.

```
Task: take a yellow box and put it into the fridge.

(1) Go to, living room(success)
(2) Pick, yellow box(success)
(3) Go to, kitchen(success)
(4) Open, fridge(fail)
(5) Put, yellow box, counter(fail)
(6) Put, yellow box, table(fail)
(7) Go to, fridge(success)
(8) Open, fridge(fail)
(9) Put, yellow box, counter(fail)
(10) Open, fridge(fail)
(11) Put, yellow box, counter(fail)
```

```
(12) Open , fridge(fail)
(13) Put , yellow box , counter(fail)
(14) Open , fridge(fail)
(15) Put , yellow box , counter(fail)
(16) Open , fridge(fail)
(17) Put , yellow box , counter(fail)
(18) Open , fridge(fail)
(19) Put , yellow box , counter(fail)
```

**Case 4: Meaningless Outputs**

Here we show a case of Qwen2-VL (Wang et al., 2024b) generating a large amount of meaningless outputs during the inference. As we can see, even though all subtasks are successful, the agent keep circling in place without making progress. These meaningless outputs quickly consume the remaining execution steps, ultimately causing the task to fail.

```
Task: find two bowls in the room and put them in the kitchen sink
(1) Go to, kitchen(success)
(2) Go to, kitchen(success)
(3) Go to, kitchen(success)
(4) Go to, kitchen(success)
(5) Go to, kitchen(success)
(6) Go to, kitchen(success)
(7) Go to, living room(success)
(8) Go to, living room(success)
(9) Go to, kitchen(success)
(10) Go to, kitchen(success)
(11) Go to, living room(success)
(12) Go to, living room(success)
(13) Go to, living room(success)
(14) Go to, living room(success)
(15) Go to, living room(success)
(16) Go to, living room(success)
(17) Go to, living room(success)
(18) Go to, living room(success)
(19) Go to, living room(success)
```

**Case 5: Solvable D1 Error**

Here we show how $D1$ error is solvable during the inference of Gemini-1.5-Pro (Team et al., 2024a). As we can see in step3, after a $D1$ error happens, a *Go to* action can effectively solve it and facilitate the success of the whole trajectory.

```
Task: fetch one crack box and one sugar box and put them on the beige table

(1) [Go to, beige_table](success)
(2) [Pick, cracker_box](fail)
(3) [Go to, cracker_box](success)
(4) [Pick, cracker_box](success)
(5) [Go to, beige_table](success)
(6) [Put, cracker_box, beige_table](success)
(7) [Go to, brown_table](success)
(8) [Put, sugar_box, brown_table](fail)
(9) [Go to, beige_table](success)
(10) [Put, sugar_box, beige_table](fail)
(11) [Go to, sugar_box](success)
(12) [Pick, sugar_box](success)
(13) [Go to, beige_table](success)
(14) [Put, sugar_box, beige_table](success)
```

```
(15) [End](success)
```

## J  Notations and Definitions

Table J8: Supplementary explanations for part symbols in the main paper.

| Symbol | Definition | Description |
|---|---|---|
| ***Evaluation Metrics*** | | |
| $T$ | Task | Task instructions in EMMOE-100, text format. |
| $k_i$ | $i$-th Keypath | An ordered chain of key subtasks for task completion. Each keypath is manually written and remains unchanged, text format (see example in Appendix C). |
| $K_T$ | Keypath Set | Composed of all keypaths of task $T$, with each keypath corresponding to a different way to complete the task. $K_T$ is also predefined and remains unchanged. |
| $k_i^{\text{check}}$ | Matched $k_i$ | We perform an ordered match between a fixed execution trajectory and $k_i$. The successfully matched portion forms $k_i^{\text{check}}$. |
| $t$ | Execution Trajectory | All planning information during the real execution process, text format. |
| replan | Re-plan | the next action that agent takes after the previous action failed, essentially belongs to a subtask, text format. |
| ***HLP Inputs*** | | |
| $I$ | Input Instruction | Multi-modal input: $\{o_{1\sim4}, s, T, inv, h, f\}$, generated automatically by HLP. |
| $o_{1\sim4}$ | Observations | Visual observations from four cardinal directions captured by the agent, image format in 256*256 resolution. |
| $s$ | System Info | Constant background information reminding the agent of its role, text format. |
| $inv$ | Inventory | Items currently held by the agent, update based on $f$, text format. |
| $h$ | Execution History | Ordered log of subtasks and their results, generated automatically by HLP, text format. |
| $f$ | Feedback | Result and error message of the last execution, generated automatically by LLE, text format. |
| ***HLP Outputs*** | | |
| $M$ | Planner Model | Fine-tuned large multimodal model for task planning. |
| $O$ | Planner Output | $O = M(I) = \{A, S, m\}$. JSON output containing analysis, subtask, and model choice, generated by M, text format. |
| $A$ | Analysis | Model's reasoning summary before generating the next subtask, text format. |
| $S$ | Subtask | $S = \{action, target\}$, next subtask with a specific action and its target, text format. |
| $m$ | Low-level Model | the execution model used to execute $S$, text format. |
| action | Action | Must be chosen from the six predefined skills, text format. |
| target | Target | The target operand or the location, text format. |

