# OpenReview forum: "EMMOE: A Comprehensive Benchmark for Embodied Mobile Manipulation in Open Environments"
_TMLR — Rejected by TMLR_

### Review · Reviewer_zAxa · 2025-10-23

**Summary Of Contributions:**

The paper introduces EMMOE (Embodied Mobile Manipulation in Open Environments), a new benchmark designed to evaluate autonomous agents on long-horizon, language-driven household tasks.

The core contribution is the benchmark itself, which integrates high-level task planning (interpreting language instructions) with low-level mobile manipulation (executing actions in a continuous 3D space). The authors collected a new dataset of 100 complex, everyday tasks, featuring detailed annotations, re-planning scenarios after failures, and diverse task types (e.g., logical, open-ended).

From this dataset, they created Supervised Fine-Tuning (SFT) and Direct Preference Optimization (DPO) sub-datasets specifically formatted to align Large Language Models (LLMs) with the data and grounding problems of embodied robotics

**Audience:**

Yes

**Audience Explanation:**

The work directly tackles a significant gap between current benchmarks , which are often isolated or simplified, and the goal of creating a general-purpose household robot. The introduction of metrics like Task Progress (TP) is a strong contribution. It allows for a more flexible and realistic evaluation of complex tasks where multiple "correct" paths exist , and it avoids rigid reliance on predefined PDDL goal states. The creation and release of SFT and DPO-formatted datasets is a practical asset for the research community , as it directly addresses the challenge of LLM-to-robot data incompatibility.

However, some weaknesses need to be addressed:

The entire framework is developed and tested exclusively in the Habitat simulator. This is a major limitation, as it provides no evidence of "sim-to-real" transfer, which remains one of the most difficult challenges in robotics.
he core EMMOE-100 dataset consists of only 100 tasks. This is a very small number to claim a "comprehensive benchmark" for training and evaluating large, general-purpose models. The SFT and DPO datasets are merely augmentations of this tiny base set (e.g., using GPT-40 to rewrite prompts), not new, diverse data.

The paper's own results demonstrate that their key method, DPO alignment, fails to generalize. The SFT+DPO model performs significantly worse than the SFT-only model on the unseen test split across most metrics (SR, PLWSR, TP, SRR). This finding undermines the value of one of their primary technical contributions.

The success rates (SR) across all models, including the authors' proposed HOMIEBOT (30.30% SR) and powerful baselines like GPT-40 (13.33% SR), are exceptionally low. This suggests the benchmark, while challenging, may be too difficult or that the current agent-based approaches are fundamentally inadequate for this level of task.

The benchmark is constrained by the six fundamental skills supported by the simulator. This limits the "everyday tasks" to simple "Go, Pick, Put, Open, Close" sequences, which is a significant simplification of real-world household manipulation.

**Broader Impact Concerns:**

Yes, a Broader Impact Statement (BIS) is present in the submission . However, it is not sufficient.

The statement is generic and fails to adequately address the specific, high-stakes ethical implications of embodied mobile manipulation in human environments. The concerns that are not sufficiently addressed are:

1. Omission of Physical Safety and Failure Risks
The BIS acknowledges "risks of misuse"  but completely omits the most direct and obvious risk of this work: physical harm from agent failure. The paper's explicit goal is a physical robot operating in a household. The submission's own results show catastrophic failure rates (e.g., a 13.33% success rate for GPT-40 and a 22.45% success rate for the "Pick" skill ). The BIS fails to discuss the foreseeable consequences of these failures in the real world (e.g., an agent misinterpreting a command, dropping a heavy or sharp object, or colliding with a human or pet). This is a critical omission for a paper focused on embodied intelligence.

2. Vague and Unsupported Claims of Bias Mitigation
The BIS acknowledges potential biases (gender, race) from "automated data generation" and claims to have "implemented measures to assess and mitigate them". This is insufficient for two reasons:
  2.a. Unsupported Claim: The paper provides zero details on what these mitigation measures are. This is an unsubstantiated claim.
  2.b. Misidentified Source of Bias: The primary bias risk is not just the augmented data (from GPT-40) , but the source data: the EMMOE-100 dataset and the Replica simulation environments. The BIS fails to analyze the potential cultural and socioeconomic biases embedded in these environments (e.g., are they all affluent, Western-style homes?) or in the selection of 100 "everyday tasks."

3. Omission of Socioeconomic Impact
The paper's explicit goal is "developing autonomous household robots". This technology has direct, profound, and foreseeable socioeconomic implications for the domestic labor market. The BIS completely fails to acknowledge or discuss this.

4. Insufficient Analysis of Long-Term Data Privacy
The BIS states that the current data is anonymized. This sidesteps the real ethical issue. The proposed agent (HOMIEBOT) is designed to operate in the real world, using four first-person-view cameras as input. The long-term application of this technology involves 24/7 video and sensor data collection from inside private homes. The BIS does not sufficiently address the severe, long-term privacy implications of this intended use case.

**Claims And Evidence:**

No

**Claims Explanation:**

While the paper's evidence is clearly presented and appears accurate, it fails to support, and in fact contradicts, one of its most critical claims: the effectiveness of its proposed training methodology.
Some of the unsupported claims are given below:

1. The proposed SFT and DPO alignment method is effective. -- This claim is directly contradicted by the paper's own results in Table 3. While the SFT+DPO model shows the best results on the training split (SR 31.84), its performance collapses on the unseen test split (SR 16.67). The simpler HOMIEBOT (SFT) baseline achieves a higher test SR (20.00), higher TP (51.19 vs. 43.39), and higher SRR (6.55 vs. 3.08). This is clear evidence of overfitting. The DPO alignment, a key contribution, failed to generalize and actively harmed the model's robustness. The authors explicitly concede this, stating, "the DPO method introduces certain generalization issues"

2.EMMOE is a "Comprehensive Benchmark." -- The dataset, EMMOE-100, contains only 100 tasks. This scale is extremely small for a benchmark intended to train and evaluate large, general-purpose models. While the data augmentation (e.g., using GPT-40 to rewrite prompts) increases data volume , it does not increase task diversity or comprehensiveness. The evidence supports it as a novel benchmark, but not a comprehensive one.

3. The low success rate is primarily due to the difficulty of EMMOE tasks. This is only partially supported. The error analysis in Table G7 shows that "Pick" actions fail approximately 78% of the time (1 - 22.45% SR). This, combined with the low overall SR of all models, suggests the Low-Level Execution (LLE) models are a fundamental bottleneck, potentially more so than the task's planning difficulty.

**Requested Changes:**

The below changes are critical for acceptance:

1. Re-evaluate and Re-frame the DPO Contribution: The paper's own results in Table 3 demonstrate that the SFT+DPO model (SR 16.67) performs worse than the SFT-only model (SR 20.00) on the unseen test split. This indicates the DPO alignment, a key proposed method, fails to generalize and instead overfits to the training data. The authors must critically address this. The claim that DPO is an effective technique for this problem is currently unsupported. The paper must be revised to either: (a) provide new results (e.g., with different hyperparameters or data) that demonstrate successful generalization, or (b) be re-framed to present this as a negative result and include a deep analysis of why DPO fails in this context. Simply noting the "generalization issues"  is insufficient

2. Remove Unsupported "Comprehensive" Claim: The title and abstract describe the benchmark as "Comprehensive". This claim is not supported by the evidence. The core dataset, EMMOE-100, contains only 100 tasks. This scale is insufficient to be considered comprehensive for training or evaluating general-purpose, large-scale models. All claims of the benchmark being "comprehensive" must be removed and replaced with more accurate, modest language (e.g., "A Novel Benchmark" or "A Pilot Benchmark").

3. Qualify the Scope of "Everyday Tasks": The paper claims to evaluate "everyday tasks" , but the action space is limited to six fundamental skills (Go to, Pick, Put, Open, Close, End). This is a significant simplification. The authors must explicitly qualify what "everyday tasks" means within this highly constrained action space, both in the abstract and introduction.

4. Provide Deeper Analysis of HLP Failures: The error analysis in Figure 3 is a strength, identifying F2 errors (grounding failures/hallucinations) as the dominant failure mode for baselines. The paper would be strengthened by a qualitative analysis of why these HLP failures occur. For example, are planners failing to understand the visual context, or are they failing to decompose the long-horizon natural language instructions (e.g., the conditional logic in the Figure 1 task )?

5. Provide a More Specific Sim-to-Real Gap Analysis: The authors correctly identify the simulation-only evaluation as a limitation. This section would be much stronger if it moved beyond a general justification for simulation and provided a specific discussion of the anticipated sim-to-real gaps for EMMOE. Given the "Pick" skill has only a 22.45% success rate in simulation, the authors should discuss which specific real-world factors (e.g., lighting, object textures, grasp physics) would be most challenging for their framework

---

> ### Author Response · Authors · 2025-11-20
>
> Thank you for your constructive suggestions and recognition of our work! We have carefully considered your advice and would like to provide the following clarifications.
>
> ---
> **Q1: Re-evaluate and Re-frame the DPO Contribution: The DPO alignment fails to generalize and instead overfits to the training data. The claim that DPO is an effective technique for this problem is currently unsupported.**
>
> - We recommend a brief review of principles and motivations of the DPO algorithm. DPO was originally designed to simplify the RLHF training pipeline (such as PPO) by omitting the sampling and reward model stages, instead directly training the final policy from preference data. It significantly reduces the difficulty and hardware requirements of RLHF training, while improving training stability and avoiding potential risks associated with reward design. In contrast, as an offline method, DPO does come with trade-offs, such as reducing generalization capabilities and relying on the base model. Therefore, we cannot say that DPO fails in the current context, as these limitations are part of its known characteristics, and there is already extensive discussions on this topic.
> - We still consider DPO to be effective because it improves evaluation metrics and success rate on the training set. Consequently, its utility should not be completely dismissed solely based on generalization issues. For example, in robotic tasks where policies are trained using RL methods, the primary focus is often on the success rate for known tasks rather than overemphasizing generalization. We believe the different focus points across domains might be the source of this misunderstanding.
>
>
>
> ---
> **Q2: Remove Unsupported "Comprehensive" Claim: The title and abstract describe the benchmark as "Comprehensive". This claim is not supported by the evidence. The core dataset, EMMOE-100, contains only 100 tasks. This scale is insufficient to be considered comprehensive for training or evaluating general-purpose, large-scale models. All claims of the benchmark being "comprehensive" must be removed and replaced with more accurate, modest language (e.g., "A Novel Benchmark" or "A Pilot Benchmark").**
>
> - We use "Comprehensive" to primarily describes the benchmark's capability. It is designed to evaluate both high-level planning models and low-level execution models, supported by a sophisticated agent system and novel evaluation metrics.
> - Besides, the benchmark includes 100 different robotic tasks, rather than 100 robot trajectories. This scale is already sufficient enough in robotics. Even the well-known Behavior series starts with only 100 tasks, let alone our tasks are more complex (with richer task properties, longer horizons, and requiring actual robotic execution). Moreover, as stated in Section 3.1, our problem definition goes beyond mere reasoning and planning.
>
>
> ---
>
> **Q3: Qualify the Scope of "Everyday Tasks": The paper claims to evaluate "everyday tasks", but the action space is limited to six fundamental skills (Go to, Pick, Put, Open, Close, End). This is a significant simplification. The authors must explicitly qualify what "everyday tasks" means within this highly constrained action space, both in the abstract and introduction.**
>
> Everyday tasks or household tasks actually constitute an extremely broad and ambiguous category, thus are hard to define. Our approach is to decompose these complex and long-horizon everyday tasks into task chains composed of fundamental primitive skills. In other words, these skills represent basic capabilities of the robot. Based on these capabilities, we can define what kinds of tasks the robot can accomplish. In Appendix, we provide a detailed list of all tasks, where you can find many composite skills such as freezing items, preparing food, cleaning tables, rearranging sofas and so on. Since primitive skills can be combined into countless composite skills, it is infeasible to define a unified scope for every possible combination. Therefore, we focus on demonstrating the scope of the robot's atomic capabilities instead.

---

> > ### Author Response · Authors · 2025-11-20
> >
> > **Q4: Provide Deeper Analysis of HLP Failures: The error analysis in Figure 3 is a strength, identifying F2 errors (grounding failures/hallucinations) as the dominant failure mode for baselines. The paper would be strengthened by a qualitative analysis of why these HLP failures occur. For example, are planners failing to understand the visual context, or are they failing to decompose the long-horizon natural language instructions (e.g., the conditional logic in the Figure 1 task )?**
> >
> > Thank you for your suggestion! In fact, we have already provided these analyses in Appendix H (Case Study). We included the original HLP trajectories and analyzed five different cases, with Case 2 specifically focusing on grounding failures. Since these contents (especially the output examples) are not suitable for the main text, we chose to present only the conclusions in Section, while placing details in the appendix for readers to review.
> >
> > ---
> >
> > **Q5: Provide a More Specific Sim-to-Real Gap Analysis: The authors correctly identify the simulation-only evaluation as a limitation. This section would be much stronger if it moved beyond a general justification for simulation and provided a specific discussion of the anticipated sim-to-real gaps for EMMOE. Given the "Pick" skill has only a 22.45% success rate in simulation, the authors should discuss which specific real-world factors (e.g., lighting, object textures, grasp physics) would be most challenging for their framework**
> >
> > Thanks for your suggestions! We'll add following contents into the limitation section:
> > - Since the simulator is not exactly the same as the real world, some unrealistic phenomena will occur during task execution. For example, the robot arm may shake drastically when a collision occurs, or the robot may directly enter the interior of the object. Besides, real world environments could not be fully realized in simulation, and the physical properties of objects in the simulation also differ from those in the real world. As a result, some operations that are feasible in the real world cannot be executed in simulation, policies trained in simulation may not generalize well to more complex scenarios in the real world. Finally, the design of the Fetch robot also affects task execution. Since its manipulator cannot reach the ground, if an object is accidentally dropped during the picking or moving process, it can no longer be picked up again. This is also one of the reasons why the success rate of Pick is so low.
> >
> > ---
> > **Q6: Broader Impact Statement (BIS) is generic and fails to adequately address the specific, high-stakes ethical implications of embodied mobile manipulation in human environments. The concerns that are not sufficiently addressed are: 1) Omission of Physical Safety and Failure Risks. 2) Vague and Unsupported Claims of Bias Mitigation. 3) Omission of Socioeconomic Impact. 4) Insufficient Analysis of Long-Term Data Privacy**
> >
> > Thanks for your suggestions! We'll rewrite BIS as following:
> > - This research utilizes publicly accessible models and simulators, ensuring that all data is anonymized and compliant with privacy regulations. However, we recognize the ethical implications of deploying such agents in human environments. Physical safety remains the primary concern; current low success rates highlight significant risks, such as object drops or collisions. As all experiments were conducted in simulation, any future real-world deployment will require strict hardware safeguards, compliant control mechanisms, and human supervision. We also acknowledge the limitations of our experimental setting: relying solely on the Replica environment may not fully represent the diversity of real-world household settings. Furthermore, the socioeconomic impact of household automation is substantial. While our goal is to enhance accessibility and alleviate domestic burdens, we advocate for responsible deployment aligned with social equity and fair labor transitions. Finally, regarding privacy, future applications must employ on-device processing and consent-based data governance to prevent misuse. To support accountability and reproducibility, all code and models will be openly shared.

---

### Review · Reviewer_KPr8 · 2025-11-05

**Summary Of Contributions:**

The paper’s contribution lies in creating the first unified benchmark (EMMOE) that connects language reasoning, spatial grounding, and embodied control, accompanied by a well-annotated dataset (EMMOE-100), novel evaluation metrics, and a hierarchical LLM-based agent (HomieBot) that demonstrates improved performance.

Strengths:
1. The paper proposes EMMOE, the first benchmark that unifies high-level task planning and low-level manipulation/navigation within a continuous-space environment.
2. The paper manually constructs EMMOE-100, a dataset of 100 complex household tasks executed in simulation using Fetch robots.
3. the paper introduces three new evaluation metrics: TP, SER, and SRR. These metrics enable more nuanced and realistic evaluation.
4. The paper designs HomieBot, a hierarchical embodied agent composed of HLP and LLE. And this paper provides comprehensive experimental evaluation.

Weaknesses:
1. The main comparisons involve huge closed-source models (GPT-4o, Gemini, o1) versus a fine-tuned 7B model. While this is informative, it does not establish whether HomieBot’s advantage comes from the dataset or from DPO alignment.
2. While the proposed TP/SER/SRR metrics are intuitively appealing, their mathematical robustness, reproducibility, and statistical behavior are not analyzed.
3. The benchmark design and system engineering are strong, but the theoretical discussion is thin.

**Audience:**

Yes

**Audience Explanation:**

This paper introduces the first benchmark that unifies high-level task planning and low-level manipulation/navigation within a continuous-space environment.

**Claims And Evidence:**

Yes

**Claims Explanation:**

Claim1: EMMOE unifies high-level and low-level embodied tasks.

The paper explicitly describes how EMMOE integrates task planning, decision making, navigation, and manipulation within continuous environments. The detailed architecture of HomieBot (Fig. 2) and the description of both High-Level Planning (HLP) and Low-Level Execution (LLE) support this claim clearly.

Claim2: EMMOE-100 provides diverse, realistic, and reasoning-rich household tasks.

Table 1 compares EMMOE-100 with other benchmarks (ALFRED, BEHAVIOR-1K, OVMM, etc.) across attributes such as task planning, re-planning, annotations, and DPO subset availability. The dataset design section also lists five types of task attributes (short-horizon, long-horizon, open-ended, logical, human-style) and includes reasoning process annotations.

Claim3: New metrics (TP, SER, SRR) provide more comprehensive evaluations.

The paper introduces mathematical definitions (Eq. 1–3) for each metric and explains how they capture execution progress, termination judgment, and adaptation ability. These are later used in experiments and yield differentiable results among models (Table 2 & 3).

**Requested Changes:**

1. A fairer comparison would include open-source models of similar scale fine-tuned on the same data.
2. A short experiment or statistical analysis showing why these metrics capture performance differences better than traditional ones.

---

> ### Author Response · Authors · 2025-11-20
>
> Thank you for your constructive suggestions and recognition of our work! We have carefully considered your advice and would like to provide the following clarifications.
>
> ---
> **Q1: The main comparisons involve huge closed-source models (GPT-4o, Gemini, o1) versus a fine-tuned 7B model. While this is informative, it does not establish whether HomieBot's advantage comes from the dataset or from DPO alignment.**
>
> First of all, we have already selected two open-sourced 7B models(Qwen2-VL-7B and MiniCPM-V2.6) for comparisons. Furthermore, we provide both the SFT and DPO version models, as well their performances, which can respectively reflect the benefits brought by the dataset and the DPO alignment.
>
> ---
> **Q2: While the proposed TP/SER/SRR metrics are intuitively appealing, their mathematical robustness, reproducibility, and statistical behavior are not analyzed.**
>
> We can fully ensure the feasibility and reproducibility of our metrics. We have provided calculation examples in Appendix, computation codes in supplementary materials. We will also release all codes on GitHub in the future.
>
>
> ---
> **Q3: The benchmark design and system engineering are strong, but the theoretical discussion is thin.**
>
> Our work mainly focuses on systematical and technical contributions, which aligns closely with the principles of TMLR. We provide comprehensive technical and experimental analyses, including baseline result analysis, error analysis, diversity testing, and case studies.
>
>
> ---
> **Q4: A short experiment or statistical analysis showing why these metrics capture performance differences better than traditional ones.**
>
> The new metrics are a complement to the traditional metrics, designed to evaluate different aspects of the agent's performance rather than to optimize measurements of the same aspect, thus making numerical comparisons difficult. For example, at the beginning of Section 3.3, we provide a detailed explanation of the differences between TP and GC.

---

### Review · Reviewer_AK1E · 2025-11-07

**Summary Of Contributions:**

This paper studies the use of large language models (LLMs) for planning in robotics, with two main contributions. First, the authors introduce the Embodied Mobile Manipulation in Open Environments (EMMOE) benchmark, which tests the ability of robotic agents to interpret user instructions and execute the task described in these instructions. A subset of this benchmark, which they call EMMOE-100, includes manually-controlled execution data for supervised fine-tuning (SFT), preference-labelled data for direct preference optimization (DPO), and other detailed annotations, such as detailed reasoning processes. They propose several evaluation metrics, including task progress (TP), which measures how close an agent is to completing the task, success end rate (SER), which measures whether the agent successfully terminates after completing the task, and success re-plan rate (SRR), which measures how well the agent adapts to new environments.

Second, the authors design the HomieBot agent for performing these tasks. This agent includes a large multimodal model, specifically LLaVA, for high level planning. The high-level planner is given four first-person view images, system information and task information in textual format, an execution history and feedback from previous executions, and an inventory of items that are currently held by the agent, and generates a JSON output that includes the next subtask to perform and an indicator of which low-level model to use for that subtask. Then, a low-level executor converts the subtask and inventory into precise low-level instructions.

Using EMMOE-100, they benchmark their model against open-source LMMs such as Qwen2-BL-7B and o1. They report their new evaluation metrics, as well as standard evaluation metrics such as success rate (SR) and Path Length Weighted Success Rate (PLWSR), showing that their model with SFT and DPO outperforms the baselines on all metrics except success re-plan rate (SRR), where o1 performs best. They also present a more fine-grained analysis of the errors, dividing the errors into logical errors (e.g. attempting to put down an object that the agent is not holding), distance errors (e.g., standing too far to reach a target), format errors (e.g. outputting an action that is not available), and execution errors (e.g. from limited capabilities of the low-level model). They find that formatting errors, specifically those related to model hallucinations or a lack of physical grounding, are the most common type of errors for the baseline models, whereas both versions of HomieBot (with and without DPO) are less prone to such errors.


# Strengths
1. The contributions are fairly extensive and significant: the EMMOE-100 dataset is very rich and requires significant human effort, while the HomieBot significantly improves over baselines.
2. The evaluation metrics and error breakdown are very informative, and shows a lot of attention to detail that will be required to make advances in LLM-based robotics.
3. The motivation is well-explained and the paper is logically structured.

# Weaknesses
1. **Related work:** The presentation of the related work could use major improvements. Currently, the related work reads like a list, with very little connection between sentences. The goal of a related works section should not be to simply give as many citations as possible and "cover your bases", but to actually situate the work within the literature. I appreciate the effort to give credit to other researchers, but this goal can be accomplished in a way that also benefits the reader. I suggest moving several of these works to the appendix, where they can be compared at the length they deserve, and using the related works section in the main paper to focus more deeply on the most relevant prior work. In its current form, I can barely read this section, much less extract the key takeaways.
2. **Lack of mathematical clarity:** As a more mathematically-inclined reader, I felt that many descriptions were too vague, which hinders the reproducibility of the paper and distracts from reading. The paper would benefit from following more of the standard conventions for mathematical communication. For example, in Section 4.2, where the format of the input instruction is introduced, the authors should specifically describe the space in which each component of $I$ lives. I understood that $o_{1\sim 4}$ denoted 4 images, and that $s$ and $T$ were text. However, the form of $f$, $inv$, and $h$ was unclear. This difficulty was present throughout the paper. As another example, in Section 3.3, terms such as $k_i$ and $K_T$ were only introduced after they were used in equations, and then only in vague terms. I suggest the authors add a subsection (or several subsections, depending on preference) to precisely establish all of their notation.
3. **Unclear train/test split in experiments:** Are the results in Table 2 for the training data? Comparing to the results in Table 3, it appears that the results for HomieBot in Table 2 are on training data. If this is the case, then this is a flaw in the presentation of the results: when comparing against the baselines, the authors should also report the results for test data.

**Audience:**

Yes

**Audience Explanation:**

Yes, the benchmark, evaluation metrics, and HomieBot agent would all be of interest to the TMLR audience.

**Broader Impact Concerns:**

A broader impact statement is already present and addresses any concerns.

**Claims And Evidence:**

No

**Claims Explanation:**

Most claims are supported, but the results in Table 2 may be misleading if they are indeed on the training data (see Weakness 3).

**Requested Changes:**

1. **Related work:** Please improve the presentation of the related work (see Weakness 1). This point is critical for securing my recommendation for acceptance.
2. **Mathematical clarity:** Please follow more standard conventions for mathematical communication (see Weakness 2). Some effort on this point would be needed for me to recommend acceptance, though I would not need the authors to make drastic changes - I recognize that the intended audience is likely not as concerned with this point as I am.
3. **Discuss/modify train/test split in Table 2:** Please clarify my concerns about Table 2 (see Weakness 3) and if needed, change Table 2 to report the metrics on the test data.

---

> ### Author Response · Authors · 2025-11-20
>
> Thank you for your constructive suggestions and recognition of our work! We have carefully considered your advice and would like to provide the following clarifications.
>
> ---
> **Q1: The presentation of the related work could use major improvements. Currently, the related work reads like a list, with very little connection between sentences.**
>
> Thank you for your suggestion! Considering that we've also discussed the background in the introduction, we rewrite and refine the related work section and move the full references to Appendix A, we also update the PDF for you to check.
>
> ---
> **Q2: Lack of mathematical clarity: The paper would benefit from following more of the standard conventions for mathematical communication.**
>
> We sincerely apologize for the lack of clarity in our writing. We prefer to introduce the formulas first and then define the notations, but now we realize that we failed to detailedly explain each symbol. Therefore, we provide detailed modality information and acquisition method for each symbol in Appendix J.
>
> ---
> **Q3: Unclear train/test split in experiments: Are the results in Table 2 for the training data? Comparing to the results in Table 3, it appears that the results for HomieBot in Table 2 are on training data. If this is the case, then this is a flaw in the presentation of the results: when comparing against the baselines, the authors should also report the results for test data.**
>
> In Section 5.3, we describe the dataset splits: ``All tasks in EMMOE-100 will be used for evaluation, and the remaining ten untrained tasks will serve as our test set``. Therefore, the dataset used in Table 2 combines train and test split, which aims to assess the model's overall capability across EMMOE-100 tasks. In contrast, Table 3 separates the train and test split, primarily to examine how different training methods affect the model's performance.

---

> > ### Comment · Reviewer_AK1E · 2025-12-05
> >
> > Thank you for addressing my concerns - the related work section in the main body of the paper is now much more readable, and the appendix section on the notation is helpful.
> >
> > A few minor remaining comments:
> > 1. Based on searching for "Appendix" in the main paper, it appears that the main paper does not contain references to Appendices A, D, G, or J. All sections of the appendix should be referenced in the main work so that the typical reader will be aware of their existence and contents. Please add these references in the appropriate sections.
> > 2. Thanks for clarifying the results for Table 2. I still suggest that the caption for Table 2 should mention that the numbers are combined over both the train and test splits - many readers may first read through the tables, and it's easy to miss the single line about this point in Section 5.3 or to not understand this point even after reading it.

---

> > > ### Author Response · Authors · 2025-12-18
> > >
> > > Thanks for your suggestions! In following revisions, we will explicitly indicate references to the appendix in the main body, and we will also add more detailed descriptions in the captions of the figures and tables.

---

### Review · Reviewer_WbRn · 2025-11-10

**Summary Of Contributions:**

The paper identifies a gap in the current embodied reasoning evaluation benchmark, which is the lack of unified evaluation of high level planning and low level control. Specifically, it points out inadequate evaluation metrics for these long, complex tasks, and the incompatibility between datasets for llm, vlm as well as vla. They trained with around 12k augmented SFT and DPO data on Video-LLaVA-7B and obtained better results comparing to sota llms.

Strength:
1. The proposed benchmark and evaluation metrics are important to tackle long horizon embodied reasoning tasks that incorporate fine grained controls.
2. The error analysis is comprehensive and insightful.


Weakness:
1. The contribution and novelty is limited here besides the proposed benchmark. The architecture and pipeline (LMM planner + skill library + sim evaluation) has numerous previous related work. Evaluation metrics are not novel as well: checking credit assignment and partial results are not under explored in past literature.
2. It seems that with the proper setup one can run RL directly, with defined metrics as reward functions. I would like to see the results of this on top of SFT/DPO.

**Audience:**

Yes

**Audience Explanation:**

It's an interesting benchmark.

**Claims And Evidence:**

No

**Claims Explanation:**

See weakness above. I would be concerned about the contribution and related work section of the paper. The benchmark seems to be novel but everything else has been explored a lot for past literature

**Requested Changes:**

See weakness

---

> ### Author Response · Authors · 2025-11-20
>
> Thank you for your constructive suggestions and recognition of our work! We have carefully considered your advice and would like to provide the following clarifications.
>
> ---
> **Q1: The contribution and novelty is limited here besides the proposed benchmark. The architecture and pipeline (LMM planner + skill library + sim evaluation) has numerous previous related work. Evaluation metrics are not novel as well: checking credit assignment and partial results are not under explored in past literature.**
>
> - We build our pipeline on a more complex task: long-horizon mobile manipulation in continuous space. Moreover, our system incorporates multiple feedback adaptation and error detection mechanisms.
> - Although the concept is not so novel, our metric use natural language for evaluation rather than simulation information, which is the key distinction.
> - Our work provides comprehensive technical and experimental analyses, including baseline result analysis, error analysis, diverse testing, and case studies. This aligns well with the principles of TMLR.
>
> ---
> **Q2: It seems that with the proper setup one can run RL directly, with defined metrics as reward functions. I would like to see the results of this on top of SFT/DPO.**
>
> We have previously compared the advantages and disadvantages of DPO and RLHF(such as PPO). Essentially, DPO provides a stable and sampling-free form of reward optimization that theoretically achieves the same goal as PPO, while significantly lowering GPU hardware requirements and training complexity. Consequently, we chose DPO as our enhancement training method.

---

> > ### Comment · Reviewer_WbRn · 2025-12-05
> >
> > Thank you for addressing my comment. My first question is cleared, for the second question, what I meant for RL method is using current approaches in RLVR (rl with verifiable reward). It seems that this is the perfect set up for applying those algorithms?
> >
> > Other than that I don't have any other questions.

---

> > > ### Author Response · Authors · 2025-12-18
> > >
> > > Thanks for your suggestion! Since the main goal of our work is to propose a household benchmark, collect a relevant dataset, design new evaluation metrics, and build a executable agent system, it may not be possible to cover an extremely broad scope in every aspect. In the future work, we plan to strengthen the model training section and further validate the RLVR approach you've suggested.

---

### Decision · Action_Editor_dSwi · 2025-12-22

**Recommendation:** Reject

**Audience:**

Yes

**Audience Explanation:**

People in the TLMR community would definitely find this interesting.

**Claims And Evidence:**

No

**Claims Explanation:**

I think the framing of the paper is not in favor of it despite having reasonable results.

The main challenge: this is essentially a benchmark paper, which is good to have. The authors than also add a baseline but make no claims about it being superior and then show that it doesn't do "superior" to other methods. This is a confusing contribution for a paper. What's the point of this baseline? Why not just evaluate previous baselines and leave it at that? Why describe it as "a sophisticated agent system which integrates models at different levels, multiple error detection and adaptation mechanisms".

In terms of the benchmark itself, I echo some of the points of Reviewer zAxa. I think the key challenge are the following statements present:
- "lack of a robotic task and benchmark that are well-aligned with realistic household task"
- "EMMOE seamlessly integrates high-level and low-level embodied tasks into a unified framework, along with three new metrics for more diverse assessment"

First, the metrics are not new. Second, the household tasks are small and not very representative of "realistic household task". To echo reviwer zAxa:
"The entire framework is developed and tested exclusively in the Habitat simulator. This is a major limitation, as it provides no evidence of "sim-to-real" transfer, which remains one of the most difficult challenges in robotics. he core EMMOE-100 dataset consists of only 100 tasks. This is a very small number to claim a "comprehensive benchmark" for training and evaluating large, general-purpose models."
"The benchmark is constrained by the six fundamental skills supported by the simulator. This limits the "everyday tasks" to simple "Go, Pick, Put, Open, Close" sequences, which is a significant simplification of real-world household manipulation."

I think if this paper made more modest claims, then with the same material, it would pass the TMLR standards which are simply that claims be supported by accurate, convincing, and clear evidence. I think right now, inaccurate due to overclaiming and thus not convincing. But for a more modest claim, it could be so.

**Resubmission Of Major Revision:**

The authors may consider submitting a major revision at a later time.